# Bounding Causal Effects with Leaky Instruments

**David S. Watson**[1]     **Jordan Penn**[1]     **Lee M. Gunderson**[2]     **Gecia Bravo-Hermsdorff**[2]

**Afsaneh Mastouri**[2]                    **Ricardo Silva**[2]

[1]King's College London
[2]University College London

## Abstract

Instrumental variables (IVs) are a popular and powerful tool for estimating causal effects in the presence of unobserved confounding. However, classical approaches rely on strong assumptions such as the *exclusion criterion*, which states that instrumental effects must be entirely mediated by treatments. This assumption often fails in practice. When IV methods are improperly applied to data that do not meet the exclusion criterion, estimated causal effects may be badly biased. In this work, we propose a novel solution that provides *partial* identification in linear systems given a set of *leaky instruments*, which are allowed to violate the exclusion criterion to some limited degree. We derive a convex optimization objective that provides provably sharp bounds on the average treatment effect under some common forms of information leakage, and implement inference procedures to quantify the uncertainty of resulting estimates. We demonstrate our method in a set of experiments with simulated data, where it performs favorably against the state of the art. An accompanying R package, `leakyIV`, is available from `CRAN`.

## 1 INTRODUCTION

Estimating causal effects from observational data can be challenging when treatments are not randomly assigned. While the task is doable under some structural assumptions [Rubin, 2005, Shpitser and Pearl, 2008, Pearl, 2009], most methods require access to data on potential confounders. This access cannot be generally guaranteed, since confounding variables may be unknown or difficult to measure. One common strategy for identifying causal effects under unobserved confounding relies on instrumental variables (IVs) [Wright, 1928, Bowden and Turkington, 1984, Angrist et al.,

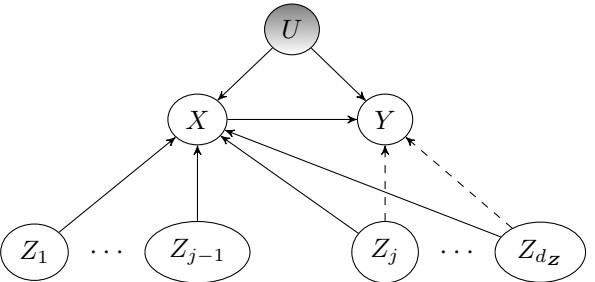

Figure 1: Causal diagram with treatment $X$, outcome $Y$, unobserved confounder $U$ (shaded), and candidate instruments $Z_1, \ldots, Z_{d_{\boldsymbol{Z}}}$. Dashed edges suggest possible violations of the exclusion criterion. Edges among $\boldsymbol{Z}$ are allowed, but omitted for simplicity.

1996], which have a direct effect on the treatment but only an indirect effect on outcomes. For instance, single nucleotide polymorphisms (SNPs) often serve as IVs in genetic epidemiology, where they may be used to investigate the impact of phenotypes (e.g., cholesterol levels) on health outcomes (e.g., cancer) in Mendelian randomization studies [Smith and Ebrahim, 2004, Didelez and Sheehan, 2007, Lawlor et al., 2008].

The IV model relies on three core conditions, formally defined in Sect. 2. Informally, we may describe IVs as variables that are (A1) *relevant*, i.e. associated with the treatment; (A2) *unconfounded*, i.e. independent of common causes between treatment and outcome; and (A3) *exclusive*, i.e. only affect outcomes through the treatment. Under some restrictions on structural equations, IVs can be used to recover causal effects despite the presence of unobserved confounding. Popular IV methods include two-stage least squares [Angrist and Imbens, 1995], as well as nonparametric extensions based on conditional moment restrictions [Newey and Powell, 2003, Newey, 2013, Bennett et al., 2019] and more recent works exploiting kernel regression [Singh et al., 2019, Muandet et al., 2020, Zhang et al., 2023] and neural networks [Hartford et al., 2017, Xu et al., 2021, Saengkyongam et al., 2022].

Of the three core conditions that characterize the IV setup, only (A1) can be immediately evaluated via observables. (A2) fails if the latent confounder between treatment and outcome is also a parent of some proposed IV(s). To continue with the Mendelian randomization example, this issue can arise when nearby variants are correlated, a phenomenon known as *linkage disequilibrium* [Reich et al., 2001]. (A3) fails if the proposed IV is a direct cause of the outcome (see Fig. 1). This can happen with complex traits in genetics, where one gene affects multiple seemingly unrelated systems through a process called *horizontal pleiotropy* [Solovieff et al., 2013]. If IV methods are naïvely applied in either case, resulting inferences can be severely biased [VanderWeele et al., 2014].

Since valid IVs may be impossible to identify *a priori*, several authors in recent years have proposed methods to estimate causal effects given just a set of candidate IVs (see Sect. 5). Details vary, but the goal is almost always to recover point estimates for the average treatment effect (ATE), possibly with associated confidence or credible intervals. We set a strictly more general target, relaxing (A3) to recover nontrivial bounds on this parameter, i.e. to *partially* identify the ATE. There is a long tradition of analytic and Bayesian methods for partial identification in IV models [Manski, 1990, Chickering and Pearl, 1996, Balke and Pearl, 1997], as well as more recent works that exploit the flexibility of stochastic gradient descent [Kallus and Zhou, 2020, Kilbertus et al., 2020, Hu et al., 2021]. Generally, a set of valid IVs is presumed—although some authors have considered the case where a single instrument is allowed to have a small effect on the outcome [Ramsahai, 2012, Conley et al., 2012, Silva and Evans, 2016].

We propose a novel procedure for bounding causal effects in settings where the exclusion criterion (A3) may not hold. Our method takes a set of *leaky instruments*, which are permitted to violate (A3) to some limited degree, and uses them to minimize confounding effects on the causal pathway of interest. Focusing on linear structural equation models (SEMs), we derive partial identifiability conditions for the ATE with access to leaky instruments, and use them to formulate a convex optimization objective. Resulting bounds are provably *sharp*—that is, they cannot be improved without further assumptions—and practically useful, providing causal information in many settings where classical methods fail. Finally, we propose a statistical test for exclusion and implement a generic bootstrapping procedure with coverage guarantees for estimated bounds.

The rest of this paper is structured as follows. We introduce the leaky IV model in Sect. 2. We present formal results in Sect. 3 and experimental results in Sect. 4. Following a review of related work in Sect. 5, we discuss limitations and generalizations of our method in Sect. 6. We conclude in Sect. 7 with a summary and future directions.

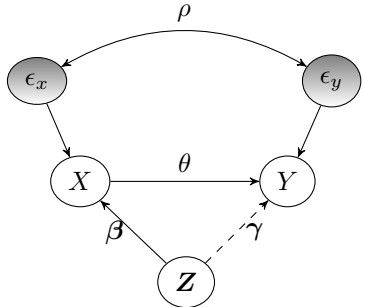

Figure 2: Causal diagram of the SEM described by Eqs. 1-3. Edge weights correspond to linear coefficients, while unobserved confounding effects are represented by the bidirected edge connecting $\epsilon_x$ and $\epsilon_y$. The dashed edge from $\boldsymbol{Z}$ to $Y$ denotes possible violations of (A3).

## 2 PROBLEM SETUP

**Notation.** We denote individual variables with capital italic letters (e.g., $X$) and bundled sets of variables in boldface capital italics (e.g., $\boldsymbol{Z} = \{Z_j\}_{j=1}^{d_{\boldsymbol{Z}}}$). We use square brackets to indicate set enumeration, e.g. $[d_{\boldsymbol{Z}}] = \{1, \ldots, d_{\boldsymbol{Z}}\}$. Parameters are symbolized as Greek letters, with boldface for vectors (e.g., $\boldsymbol{\beta}$) and boldface capitals for matrices (e.g., $\boldsymbol{\Sigma}$).

Standard notation for (co)variances can sometimes be confusing. Here we use the capital $\boldsymbol{\Sigma}$ for any such quadratic expectation, regardless of whether it is a scalar, vector, or matrix—the subscripts will contain all the necessary information as to its dimensions. For instance, we write $\boldsymbol{\Sigma}_{xy}$ for $\text{Cov}(X, Y)$ (as opposed to $\sigma_{xy}$) and $\boldsymbol{\Sigma}_{xx}$ for $\text{Var}(X)$ (as opposed to $\sigma_x^2$). As a convenient byproduct, the notation generalizes more naturally to vector-valued variables.

**The Leaky IV Setting.** Consider a linear SEM with treatment $X \in \mathbb{R}$, outcome $Y \in \mathbb{R}$, and a set of candidate IVs $\boldsymbol{Z} \in \mathbb{R}^{d_{\boldsymbol{Z}}}$. Assume that all variables have mean 0 and finite variance. Data are generated according to the following process (see Fig. 2):

$$X = \boldsymbol{\beta} \cdot \boldsymbol{Z} + \epsilon_x \qquad (1)$$

$$Y = \boldsymbol{\gamma} \cdot \boldsymbol{Z} + \theta X + \epsilon_y \qquad (2)$$

$$\boldsymbol{\Sigma}_{\epsilon\epsilon} = \begin{bmatrix} \eta_x^2 & \rho\eta_x\eta_y \\ \rho\eta_x\eta_y & \eta_y^2 \end{bmatrix}, \qquad (3)$$

where $\theta \in \mathbb{R}$ and $\boldsymbol{\beta}, \boldsymbol{\gamma} \in \mathbb{R}^{d_{\boldsymbol{Z}}}$ are linear weights; and $\epsilon_x, \epsilon_y \in \mathbb{R}$ are residuals with mean 0, standard deviations $\eta_x, \eta_y \geq 0$, and correlation $\rho \in [-1, 1]$. This latter parameter $\rho$ quantifies the magnitude and direction of unobserved confounding. To better interpret results, we assume all $Z$'s are on roughly the same scale (e.g., standardized to unit variance). We make no further assumptions about the distribution of $\boldsymbol{Z}$. Our goal is to bound $\theta$, which denotes the average treatment effect (ATE) of $X$ on $Y$.

In the classical nonparametric IV setting, we have a set of unobserved confounders $U \in \mathbb{R}^{d_U}$ with direct effects on both $X$ and $Y$. Then $Z$ is a set of *valid instruments* if and only if the following conditions are satisfied:[1]

(A1) *Relevance:* $Z \not\perp\!\!\!\perp X$
(A2) *No confounding:* $Z \perp\!\!\!\perp U$
(A3) *Exclusion criterion:* $Z \perp\!\!\!\perp Y \mid \{X, U\}$.

Adapting these assumptions to a linear SEM, we posit that $U$ is a parent of both noise variables satisfying $\epsilon_x \perp\!\!\!\perp \epsilon_y \mid U$ and equate (conditional) independence with (conditional) covariance of zero. Under these conditions, we may compute treatment effects via two-stage least squares (2SLS) [Bowden and Turkington, 1984]. For this procedure, we solve Eq. 1 with ordinary least squares (OLS) and substitute fitted values from this model for $X$ in Eq. 2, which is in turn solved via OLS. The resulting $\hat{\theta}^{\text{2SLS}}$ is our ATE estimate. Note that (A3) implies that $\|\gamma\| = 0$, since in this case each $Z_j \in Z$ receives zero weight in Eq. 2.

We relax the exclusion criterion and consider two modified variants using scalar or vector-valued thresholds.

(A3$'_s$) *Scalar $\tau$-exclusion:* $\|\gamma\|_p \leq \tau$
(A3$'_v$) *Vector $\tau$-exclusion:* $\forall j \in [d_Z] : |\gamma_j| \leq \tau_j$.

That is, we allow $Z$ to have some direct effect on $Y$ but restrict this influence either by placing an upper bound on the $L_p$-norm of the $\gamma$ coefficients (scalar-valued $\tau$) or by placing separate thresholds on the magnitude of each individual coefficient (vector-valued $\tau$). We derive sharp ATE bounds for both cases, as well as a closed form solution under (A3$'_s$) when $p = 2$.

We call variables that satisfy (A1), (A2), and either form of $\tau$-exclusion *leaky instruments*. These features are technically observed confounders (at least those with nonzero $\gamma$ coefficients). Were it not for the unobserved confounding induced by $\rho$, causal effects could be calculated by integrating over $Z$, as in the backdoor adjustment [Pearl, 2009, Ch. 3.3]. Unfortunately, this option is unavailable when $\rho \neq 0$. By exploiting known leakage threshold(s), however, we show how to recover sharp bounds on the ATE.

Unlike other methods designed to accommodate potential violations of the exclusion criterion (see Sect. 5), we do not assume that some proportion of candidate IVs are valid, or that biases introduced by direct links from $Z$ to $Y$ cancel out. On the contrary, we explicitly allow for a dense set of nonzero $\gamma$ weights, provided they satisfy some form of $\tau$-exclusion. As our experiments below demonstrate, this method naturally accommodates sparse $\gamma$ vectors without presuming them upfront.

---

[1]The "exogeneity" assumption in econometrics is sometimes equated with (A2), and sometimes with the conjunction of (A2) and (A3) (see, e.g., [Wooldridge, 2019, Ch. 15]). We avoid all talk of exogeneity to avoid confusion.

## 3 THEORY

In this section, we show how to partially identify the ATE under bounded violations of the exclusion criterion and propose methods for statistical inference.

### 3.1 SCALAR $\tau$-EXCLUSION

We begin with a scalar threshold on information leakage from $Z$ to $Y$. As a first pass, we may formalize our objective as follows:

$$\min/\max_{\beta, \gamma, \theta, \eta_x, \eta_y, \rho} \quad \theta$$
$$\text{s.t.} \quad \Sigma_{\mathcal{M}} = \Sigma,$$
$$\eta_x \geq 0, \eta_y \geq 0, -1 \leq \rho \leq 1, \|\gamma\|_p \leq \tau,$$

where $\Sigma$ is the observational covariance matrix of $\{X, Y, Z\}$ and $\Sigma_{\mathcal{M}}$ is the model covariance matrix implied by Eqs. 1, 2 and 3.

Though technically correct, this formulation is unnecessarily complex. It suggests a potentially high-dimensional constrained optimization problem that is not obviously amenable to polynomial programming techniques. To simplify it, we provisionally assume access to the population covariance matrix $\Sigma$. (We discuss methods for estimating these parameters in Sect. 3.3.) This allows us to solve directly for $\beta$ and $\eta_x^2$:

$$\beta = \Sigma_{zz}^{-1} \cdot \Sigma_{zx}, \quad \eta_x^2 = \Sigma_{xx} - \beta \cdot \Sigma_{xz}.$$

With these parameters fixed, the remaining coefficients $\theta, \gamma$ are rendered deterministic functions of $\rho$, with some special care for the non-negativity constraint on $\eta_y$. To see this, it helps to define the scalars:

$$\kappa_{xx} := \Sigma_{xx} - \Sigma_{xz} \cdot \Sigma_{zz}^{-1} \cdot \Sigma_{zx} = \eta_x^2$$
$$\kappa_{xy} := \Sigma_{xy} - \Sigma_{xz} \cdot \Sigma_{zz}^{-1} \cdot \Sigma_{zy}$$
$$\kappa_{yy} := \Sigma_{yy} - \Sigma_{yz} \cdot \Sigma_{zz}^{-1} \cdot \Sigma_{zy}$$

These terms correspond, respectively, to the conditional variance of $X$ given $Z$ ($\kappa_{xx}$), the conditional covariance of $X$ and $Y$ given $Z$ ($\kappa_{xy}$), and the conditional variance of $Y$ given $Z$ ($\kappa_{yy}$). Thus, by the Cauchy-Schwarz inequality, $\kappa_{xx}\kappa_{yy} \geq \kappa_{xy}^2$. With these definitions in hand, we are now ready to characterize the relationship between $\rho$ and $\theta$. (See Appx. A for all proofs.)

**Lemma 1** (*ATE as a function of confounding*). There is a bijective, strictly decreasing function $f : [-1, 1] \mapsto \mathbb{R}$ that maps values of the confounding coefficient $\rho$ to the ATE $\theta$:

$$\theta = f(\rho) := \kappa_{xx}^{-1}\Big(\kappa_{xy} - \sqrt{\kappa_{xx}\kappa_{yy} - \kappa_{xy}^2} \tan\big(\arcsin(\rho)\big)\Big).$$

This function takes the shape of a rotated sigmoid (see Fig. 3A). Note that when $\rho = 0$, there is no unobserved

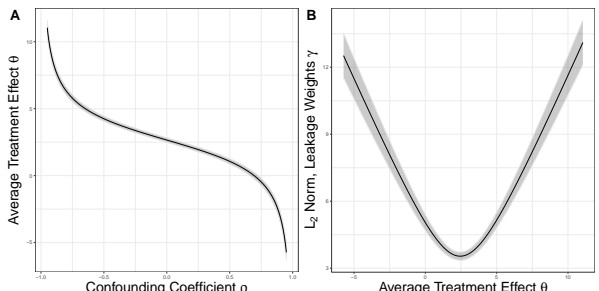

Figure 3: Example curves illustrating the relationships between parameters in the leaky IV model. **(A)** A $\rho$-$\theta$ curve maps the relationship between latent confounding and causal effects. **(B)** A $\theta$-$\|\boldsymbol{\gamma}\|_2$ curve maps the relationship between causal effects and information leakage. Shading represents 95% confidence intervals estimated via the bootstrap.

confounding and $\theta$ can simply be estimated by OLS (if $\|\boldsymbol{\gamma}\| = 0$) or backdoor adjustment on $\boldsymbol{Z}$ (if $\|\boldsymbol{\gamma}\| > 0$).

Because strong confounding induces extreme values of $\theta$, more informative bounds on the ATE can be derived if subject matter knowledge allows us to truncate the range of $\rho$. This is precisely what (A3$'_s$) achieves, although the exact form of this truncation depends on our choice of norm. To see how $\tau$-exclusion restricts $\rho$'s range, we must spell out the relationship between the ATE and leaky weights $\boldsymbol{\gamma}$.

**Lemma 2** (*Leakage as a function of ATE*). There is a surjective function $g_p : \mathbb{R} \mapsto \mathbb{R}_{\geq 0}$ that maps values of the ATE $\theta$ to the $L_p$ norm of the leakage weights $\boldsymbol{\gamma}$:

$$\|\boldsymbol{\gamma}\|_p = g_p(\theta) := \|\boldsymbol{\alpha} - \theta\boldsymbol{\beta}\|_p,$$

where $\boldsymbol{\alpha} := \boldsymbol{\Sigma}_{\boldsymbol{zz}}^{-1} \cdot \boldsymbol{\Sigma}_{\boldsymbol{zy}}$ represents the expected weights of an OLS regression of $Y$ on $\boldsymbol{Z}$.

For the special case of $p = 2$, this function is quadratic (see Fig. 3B). Recall that the $L_p$ norm is convex for all $p \geq 1$ and strictly convex for $p \in (1, \infty)$. Though everywhere differentiable for $p \in [2, \infty)$, the norm may be non-differentiable at countably many points for $p \in [1, 2)$.

The leakage threshold $\tau$ defines a feasible region of possible models. While we presume that this parameter is provided upfront (more on this in Sect. 6), it cannot be made arbitrarily small in the leaky IV setting. Specifically, the lower bound corresponds to particular values of $\theta$ and $\rho$.

**Lemma 3** (*Minimum leakage as a function of ATE*). The minimum degree of leakage consistent with the data can be obtained by solving the following linear regression task in $L_p$ space:

$$\check{\theta}_p := \operatorname*{arg\,min}_{\theta \in \mathbb{R}} \ g_p(\theta).$$

In the special case of $p = 2$, the optimum is given by the standard OLS estimator $\check{\theta}_2 = (\boldsymbol{\beta} \cdot \boldsymbol{\beta})^{-1} \boldsymbol{\beta} \cdot \boldsymbol{\alpha}$. Though

closed form solutions are not available for arbitrary $p$—even in the well-studied case of $p = 1$ [Pollard, 1991, Portnoy and Koenker, 1997, Chen et al., 2008]—the value is easily computed via numerical methods. Note that $\check{\theta}_p$ is unique for any strictly convex $L_p$ norm, but may form a compact interval for $p \in \{1, \infty\}$.

Next, we find the corresponding value(s) of $\rho$.

**Lemma 4** (*Minimum leakage as a function of confounding*). Define $h_p := g_p \circ f$, such that $h_p : [-1, 1] \mapsto \mathbb{R}_{\geq 0}$ maps values of $\rho$ to $\|\boldsymbol{\gamma}\|_p$. For any $\check{\theta}_p$ (either a unique solution or any point on the compact interval of solutions), $h_p$ achieves its minimum at:

$$\check{\rho}_p := \operatorname*{arg\,min}_{\rho \in [-1, 1]} \ h_p(\rho) = f^{-1}(\check{\theta}_p)$$

$$= \sin\left( \arctan\left( \frac{\kappa_{xy} - \check{\theta}_p \kappa_{xx}}{\sqrt{\kappa_{xx}\kappa_{yy} - \kappa_{xy}^2}} \right) \right).$$

Lemmas 3 and 4 provide an essential criterion for partial identification in the leaky IV model. Define $\check{\tau}_p := g_p(\check{\theta}_p) = h_p(\check{\rho}_p)$. (Observe that this value is unique even when $\check{\theta}_p$ and $\check{\rho}_p$ are not.) Let $\theta^*$ denote the true ATE, with corresponding leakage weights $\boldsymbol{\gamma}^* = \boldsymbol{\alpha} - \theta^*\boldsymbol{\beta}$ and oracle threshold $\tau_p^* := \|\boldsymbol{\gamma}^*\|_p$, so named because it quantifies the precise (and unidentifiable) amount of information leakage from $\boldsymbol{Z}$ to $Y$ in the true data generating process. These minimum and oracle thresholds fully characterize identifiability conditions in the leaky IV model.

**Theorem 1** (*Identifiability*). *Assume Eqs. 1, 2, and 3 and assumptions (A1), (A2), and (A3$'_s$) hold for some $p \geq 1$. Then ATE bounds are:*

- *undefined for all $\tau < \check{\tau}_p$;*
- *identifiable but invalid for all $\tau \in [\check{\tau}_p, \tau_p^*)$; and*
- *identifiable and valid for all $\tau \geq \tau_p^*$.*

*Moreover, the true ATE is identifiable iff $\tau_p^* = \check{\tau}_p$ and $g_p$ attains a unique minimum, in which case $\theta^* = \check{\theta}_p$.*

The three-partition of threshold space implied by Thm. 1 is visualized in Fig. 4 for $p = 2$, where we see how bounds go from nonexistent (grey striped region) to small but erroneous (red shaded region), only becoming valid above the oracle threshold $\tau_2^*$. Note that the perpendicular lines $\|\boldsymbol{\gamma}\|_p = \tau_p^*$ and $\theta = \theta^*$ intersect at a point on the leakage curve. This illustrates that valid bounds in the leaky IV model are not generally symmetric about $\theta^*$. In fact, for $p \in (1, \infty)$ and $\tau_p^* > \check{\tau}_p$, the true ATE $\theta^*$ will coincide with one extremum of the partial identification interval at $\tau = \tau_p^*$.

The identifiability conditions of Thm. 1 have an immediate consequence for the classic linear IV model, which is a special case of our leaky IV model with $\tau = 0$.

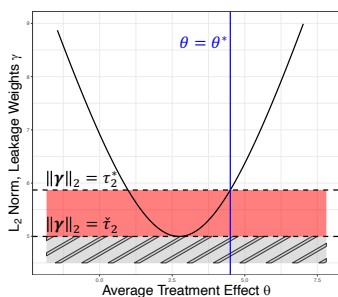

Figure 4: Minimum and oracle leakage values impose a three-partition of the threshold space. Below $\check{\tau}_2$, we have *the infeasible region* (grey striped area), where no configuration of latent parameters satisfies our structural constraints. Between $\check{\tau}_2$ and $\tau_2^*$, we have *the error region* (red area), where bounds are identifiable but invalid. Above $\tau_2^*$, we have *the valid region* (rest of the plot), where bounds are guaranteed to contain the true ATE $\theta^*$, represented by the vertical blue line.

**Corollary 1.1.** Under the assumptions of Thm. 1, the exclusion criterion (A3) holds iff $\tau_p^* = \check{\tau}_p = 0$, in which case $\theta^* = \check{\theta}_2 = \theta^{2\text{SLS}}$.

As we will see in the sequel, this constraint has falsifiable consequences that can motivate the use of the leaky IV approach in practice.

We now reformulate our optimization task:

$$\min_{\rho \in [-1,1]} / \max \quad \theta \quad \text{s.t.} \quad h_p(\rho) = \tau.$$

This is a straightforward one-dimensional objective where all structural constraints have been absorbed into a single function $h_p$. We now have all the ingredients in place to state our main result.

**Theorem 2** (*ATE bounds*). *Assume the conditions of Thm. 1 hold for some $\tau \geq \tau_p^*$. Then for any $\check{\rho}_p$ (either a unique solution or any point on the compact interval of solutions), there exist unique min/max values of the confounding coefficient consistent with the posited information leakage:*

$$\rho_{\tau,p}^- := \min_{\rho \in [-1,\check{\rho}_p]} \quad \rho \quad s.t. \quad h_p(\rho) = \tau$$
$$\rho_{\tau,p}^+ := \max_{\rho \in [\check{\rho}_p,1]} \quad \rho \quad s.t. \quad h_p(\rho) = \tau.$$

*Plugging these values into $f$ produces valid and sharp ATE bounds:*

$$\theta_{\tau,p}^- = f(\rho_{\tau,p}^+), \quad \theta_{\tau,p}^+ = f(\rho_{\tau,p}^-).$$

Analytic solutions are generally intractable for $p \neq 2$. However, Thm. 2 guarantees the existence and uniqueness of valid, sharp ATE bounds in the leaky IV model for any $p \geq 1$. These values can be readily computed with numerical methods, e.g. linear programming techniques [Bertsimas]

and Tsitsiklis, 1997]. For the $L_2$ case, we derive the following solution in closed form.

**Corollary 2.1.** Under the assumptions of Thm. 2 with $p = 2$, min/max ATE values are given by:

$$\check{\theta}_2 \pm (\boldsymbol{\beta} \cdot \boldsymbol{\beta})^{-1} \sqrt{(\boldsymbol{\beta} \cdot \boldsymbol{\beta})(\tau^2 - \boldsymbol{\alpha} \cdot \boldsymbol{\alpha}) + (\boldsymbol{\alpha} \cdot \boldsymbol{\beta})^2}.$$

### 3.2 VECTOR $\tau$-EXCLUSION

Our scalar $\tau$-exclusion criterion is somewhat crude, as it applies a single threshold on a summary statistic of all $\boldsymbol{\gamma}$ weights. In many cases, however, background knowledge may license a more fine-grained approach that applies separate thresholds either to individual candidate instruments or groups thereof. For instance, in a Mendelian randomization study, we may partition SNPs by chromosome, exploiting biological knowledge to permit more or less leakage as we move across the genome. Alternatively, we may impose the restriction that our $\boldsymbol{Z}$ variables should be more "relevant" than "leaky", with each $\beta_j$ coefficient exceeding the corresponding $\gamma_j$ in absolute value.

These considerations inspire a more heterogeneous relaxation of the exclusion criterion characterized by (A3$_v'$), which we refer to as *vector $\tau$-exclusion*. Fortunately, the solution in this case follows naturally from our previous analysis. Suppose that all thresholds are strictly positive, i.e. that $\|\boldsymbol{\tau}\|_0 = d_{\boldsymbol{Z}}$. Then we simply perform a linear transformation of all candidate instruments, scaling them by their respective leakage thresholds to create modified variables $\tilde{Z}_j := Z_j/\tau_j$. Let $\boldsymbol{\tau}^+ := [1, 1, \boldsymbol{\tau}]$ denote an augmented threshold vector of length $2 + d_{\boldsymbol{Z}}$, with dummy entries for $X$ and $Y$. Then, we define a square transformation matrix $\boldsymbol{T}$ with entries $\boldsymbol{T}_{ij} := 1/(\tau_i^+ \tau_j^+)$ and update our covariance parameters:

$$\tilde{\boldsymbol{\Sigma}} := \boldsymbol{T} \odot \boldsymbol{\Sigma},$$

where $\odot$ denotes the Hadamard product (i.e., entrywise multiplication). Plugging this matrix into the equations of Sect. 3.1 produces transformed linear weights $\tilde{\boldsymbol{\alpha}}, \tilde{\boldsymbol{\beta}}, \tilde{\boldsymbol{\gamma}}$. Vector $\tau$-exclusion can now be rewritten as:

(A3$_{v2}'$) *Vector $\tau$-exclusion:* $\|\tilde{\boldsymbol{\gamma}}\|_\infty \leq 1$.

All previous results go through just the same, including identifiability conditions (Thm. 1) and optimal ATE bounds (Thm. 2).

This strategy will need to be modified if we wish to impose $\tau_j = 0$ for some $j \in [d_{\boldsymbol{Z}}]$—i.e., to treat some variable(s) as valid IVs that satisfy the classical exclusion criterion. Let $\boldsymbol{S}_0 \subset [d_{\boldsymbol{Z}}]$ pick out all and only those features such that $\tau_j = 0$, with complementary subset $\boldsymbol{S}_1 := [d_{\boldsymbol{Z}}] \backslash \boldsymbol{S}_0$. Then for each $j \in \boldsymbol{S}_0$, we set the corresponding entry in $\boldsymbol{\tau}^+$ to 1 in our construction of the transition matrix $\boldsymbol{T}$ to avoid

division by zero, and update the formula for $\tilde{\gamma}$ to reflect the reduced degrees of freedom:

$$\tilde{\gamma}_j = \begin{cases} 0, & \text{if } j \in \boldsymbol{S}_0; \\ \tilde{\alpha}_j - \theta\tilde{\beta}_j, & \text{otherwise}, \end{cases}$$

for all $j \in [d_{\boldsymbol{Z}}]$. We modify the definitions of $g_p, h_p$, and $\tau_p^*$ to restrict their range to just those $j \in \boldsymbol{S}_1$. Now (A3$'_{v2}$) applies and produces sharp bounds, as desired.

## 3.3 INFERENCE

Thm. 2 provides an exact solution with the population covariance matrix $\boldsymbol{\Sigma}$. In practice, of course, all parameters must be estimated from finite data. We generally take the sample covariance matrix $\hat{\boldsymbol{\Sigma}}$ as our plug-in estimator, but many alternatives are possible. Numerous Bayesian [Leonard and Hsu, 1992, Daniels and Kass, 1999, Gelman et al., 2014, Ch. 3.6] and penalized likelihood [Schäfer and Strimmer, 2005, Warton, 2008, Won et al., 2012] methods have been proposed for this task, or the closely related task of estimating a regularized precision matrix [Friedman et al., 2007, Cai et al., 2011, Mazumder and Hastie, 2012]. Several of these options are implemented in our accompanying software package. These alternatives may be especially attractive in high-dimensional settings with a large number of leaky IVs to ensure a positive definite $\hat{\boldsymbol{\Sigma}}$.

We use a variety of estimators in our experiments below and augment the procedure with inference techniques. First, we describe a parametric test of the exclusion criterion, as foreshadowed by Corollary 1.1. In the linear IV model with $d_{\boldsymbol{Z}} \geq 2$, it is well known [Kuroki and Cai, 2005, Chen et al., 2014, Silva and Shimizu, 2017] that exclusion imposes a set of so-called *tetrad constraints* on covariance parameters of the form:

$$\boldsymbol{\Sigma}_{z_j y}\boldsymbol{\Sigma}_{z_k x} - \boldsymbol{\Sigma}_{z_j x}\boldsymbol{\Sigma}_{z_k y} = 0.$$

If this holds for all nonidentical pairs of candidate instruments $j, k \in [d_{\boldsymbol{Z}}]$, then $\boldsymbol{\alpha}, \boldsymbol{\beta}$ vectors are parallel and information leakage goes to zero. Define the $d_{\boldsymbol{Z}} \times 2$ matrix $\boldsymbol{\Lambda} := [\boldsymbol{\Sigma}_{zx}, \boldsymbol{\Sigma}_{zy}]$. Then our test statistic is $\psi := \det(\boldsymbol{\Lambda} \cdot \boldsymbol{\Lambda})$ and our null hypothesis is $H_0 : \psi = 0$, which is necessary and sufficient for $\check{\tau}_p = 0$.

We propose to test $H_0$ via Monte Carlo, estimating $\hat{\theta}^{\text{2SLS}}$ on the original data and creating a null covariance matrix $\boldsymbol{\Sigma}^0$ by replacing $\hat{\boldsymbol{\Sigma}}_{zy}$ with $\boldsymbol{\Sigma}_{zy}^0 := \hat{\boldsymbol{\Sigma}}_{zx}\hat{\theta}^{\text{2SLS}}$. We assume that samples are distributed according to some $P_{\boldsymbol{\Sigma}} \in \mathcal{P}$, where $\mathcal{P}$ denotes a family of distributions parameterized by a covariance matrix $\boldsymbol{\Sigma}$—obvious examples include multivariate Gaussian and $t$-distributions, although alternatives such as multivariate binomial or Poisson distributions are also viable [Krummenauer, 1998, Jiang et al., 2021]. So long as we can sample data under fixed values of $\boldsymbol{\Sigma}$, we can perform the following test.

**Theorem 3** (*Exclusion test*). *Let $\mathcal{D}_n = \{x_i, y_i, \boldsymbol{z}_i\}_{i=1}^n$ be a dataset generated according to the conditions of Thm. 1, with $\mathcal{D}_n \sim P_{\boldsymbol{\Sigma}}$, $d_{\boldsymbol{Z}} \geq 2$, and sample estimate $\hat{\psi}_n$. Construct a null covariance matrix $\boldsymbol{\Sigma}^0$ as detailed above. Draw $B$ synthetic datasets of size $n$, $\mathcal{D}_{n,(b)}^0 \sim P_{\boldsymbol{\Sigma}^0}$, and record the test statistic $\psi_{n,(b)}^0$ for all $b \in [B]$. Then as $n, B \to \infty$, the following is an asymptotically valid p-value against $H_0$:*

$$p_{\text{MC}} = \frac{\#\{b : \psi_{(b)}^0 \geq \hat{\psi}_n\} + 1}{B + 1}.$$

Thm. 3 describes a frequentist method for testing the exclusion criterion in linear IV models. Sufficiently small values of $p_{\text{MC}}$ can motivate a leaky approach, as 2SLS results in biased ATE estimates when (A3) fails. Note that $\check{\tau}_p = 0$ is a necessary but insufficient condition for exclusion, which additionally requires that $\tau_p^* = 0$. A minimum possible leakage of zero provides no evidence that the true leakage is in fact zero.

Next, we introduce a nonparametric bootstrapping procedure [Efron, 1979] to quantify the uncertainty of ATE bounds. Specifically, we draw $B$ many datasets of size $n$ by sampling with replacement from the input data and estimate the covariance matrix $\hat{\boldsymbol{\Sigma}}_{(b)}$ for each $b \in [B]$. Our target parameters are $(\theta^-, \theta^+)_{\tau,p}^*$, i.e. the bounds we would expect from a *partial identification oracle* (henceforth a $\boldsymbol{\Sigma}$-oracle) with knowledge of the population covariance matrix for observed variables $\{X, Y, \boldsymbol{Z}\}$. Note that even with access to the true $\boldsymbol{\Sigma}$, $\tau_p^*$ and $\theta^*$ remain unidentifiable, and so we distinguish between $\boldsymbol{\Sigma}$-oracles and oracles *tout court*, who are additionally omniscient with respect to latent parameters and therefore able to point identify the ATE. By contrast, with $\tau \in [\check{\tau}_p, \tau_p^*)$, a $\boldsymbol{\Sigma}$-oracle will produce invalid bounds that lie in the error region of Fig. 4 (see Thm. 1).

Since $\check{\tau}_p$ depends on the data, it is possible that some bootstraps may violate the partial identifiability criterion $\tau \geq \check{\tau}_p$, especially if the sample size is small and/or the selected threshold is close to the true leakage minimum as determined by a $\boldsymbol{\Sigma}$-oracle. Our estimator is undefined when the feasible region is empty, so we discard any offending bootstraps. As $\check{\tau}_p$ can be estimated on the full dataset upfront, this issue may be mitigated by selecting a threshold sufficiently high above this value. The procedure comes with the following coverage guarantee.

**Theorem 4** (*Coverage*). *Let $\mathcal{D}_n = \{x_i, y_i, \boldsymbol{z}_i\}_{i=1}^n$ be a dataset generated according to the conditions of Thm. 1. Draw $B$ samples with replacement from $\mathcal{D}_n$, subject to $\tau \geq \check{\tau}_{p,(b)}$ for all $b \in [B]$. For a given $\diamond \in \{-, +\}$ and level $\alpha \in (0, 1)$, we construct the confidence interval $\hat{C}_n = [\hat{q}_l, \hat{q}_u]$ as follows. Let $\hat{q}_l$ be the lth smallest value of the bootstrap distribution for $\hat{\theta}_{\tau,p}^{\diamond}$, with $l = \lceil (B+1)(\alpha/2) \rceil$. Let $\hat{q}_u$ be the uth smallest value of the same set, with $u =*

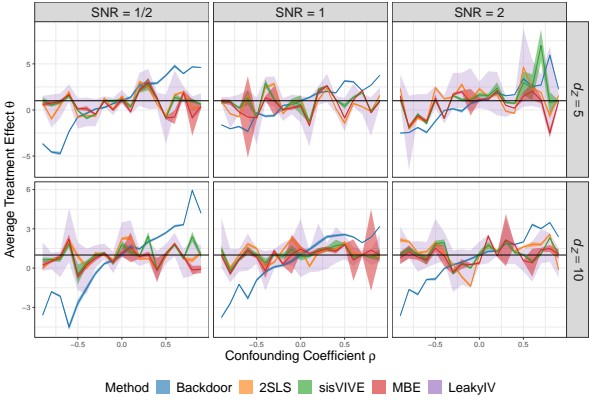

Figure 5: Comparison against various methods at a range of values for the confounding coefficient $\rho$, SNR for $Y$, and number of candidate instruments $d_{\boldsymbol{Z}}$. The horizontal black line at $\theta = 1$ represents the true ATE $\theta^*$.

$\lceil (B + 1)(1 - \alpha/2) \rceil$. *Then as* $n, B \to \infty$, *we have:*

$$\mathbb{P}\big(\theta_{\tau,p}^{\diamond *} \in \hat{C}_n\big) \geq 1 - \alpha.$$

We can smooth the bootstraps with kernel density estimation (see Sect. 4) or use a Bayesian bootstrap to get an approximate posterior distribution [Rubin, 1981]. Either way, Thm. 4 provides a template for testing claims about whether, for instance, zero lies above or below the partial identification interval with high probability.

## 4 EXPERIMENTS

For full details of all simulation experiments, see Appx. B. Code for reproducing all results and figures can be found on our dedicated GitHub repository.[2] In this section, we use $p = 2$ throughout and estimate all covariance parameters via maximum likelihood.

For our benchmark experiments, we generate data according to Eqs. 1-3 with the following process:

$$\boldsymbol{Z} \sim \mathcal{N}(0, \boldsymbol{\Sigma}_{\boldsymbol{zz}}) \quad \boldsymbol{\beta} \sim \mathcal{N}(0, 1)$$
$$\boldsymbol{\gamma} \sim \mathcal{N}(0, 1) \times \zeta \quad \epsilon_x, \epsilon_y \sim \mathcal{N}(0, \boldsymbol{\Sigma}_{\boldsymbol{\epsilon\epsilon}}),$$

and fixed $\boldsymbol{\Sigma}_{yy} = 10$. The scaling factor $\zeta$ and residual variance parameters $\eta_x^2, \eta_y^2$ are chosen to ensure that the signal-to-noise ratio (SNR) of Eqs. 1 and 2 are fixed at the desired level (for details, see Appx. B.2). We simulate data from the leaky IV model under a range of hyperparameters:

- Dimensionality $d_{\boldsymbol{Z}}$ is selected from $\{5, 10\}$.
- The covariance matrix $\boldsymbol{\Sigma}_{\boldsymbol{zz}}$ is either diagonal or Toeplitz with autocorrelation $0.5$. In either case, we set marginal variance to $1/d_{\boldsymbol{Z}}$ for each $Z$.
- The confounding coefficient $\rho$ is selected from $\{-0.9, -0.8, \ldots, 0.9\}$.

[2]https://github.com/dswatson/leakyIV.

- The SNR for $X$ is selected from $\{0.5, 1, 2\}$.
- The SNR for $Y$ is selected from $\{0.5, 1, 2\}$.

Taking the Cartesian product of all these hyperparameters generates a grid of 684 unique simulation configurations. We hold the sparsity of $\boldsymbol{\gamma}$ fixed at $0.2$ and set the true ATE $\theta^*$ to 1 across all experiments.

**Point Estimators.** We present the mean and standard deviation of ATE estimates for a range of methods, computed across 50 runs of $n = 1000$. For our own own approach, LeakyIV, we set $\tau = 1.1\tau_*^*$ and shade the interval between our mean estimates for $(\hat{\theta}^-, \hat{\theta}^+)_{\tau,2}$. We benchmark against two classic methods—the backdoor adjustment and 2SLS—as an illustrative baseline. We also compare our results to two methods designed for causal inference with some invalid instruments: sisVIVE, which performs implicit feature selection via an $L_1$ penalty on the candidate IVs [Kang et al., 2016]; and mode-based estimation (MBE), which treats invalid instruments as outliers that can be ignored using robust inference techniques [Hartwig et al., 2017]. We refer readers to the original papers for details on each.

Results for $\boldsymbol{\Sigma}_{\boldsymbol{zz}} = $ Toeplitz, $\mathrm{SNR}_X = 2$ are presented in Fig. 5. (Results are broadly similar for alternative simulations; see Appx. B.1.) We find that the backdoor adjustment is systematically biased downward for $\rho < 0$ and upward for $\rho > 0$, exactly as theory predicts. In most cases, confounding effects on either end of the $x$-axis are sufficiently strong to send the curve beyond the limits of our estimated partial identification interval. Alternative methods designed for the IV setting fare better, but still behave somewhat erratically. MBE in particular appears prone to occasional bursts of uncertainty, especially under extreme confounding.

By contrast, our bounds contain the true ATE in 683 out of 684 settings, or 99.85% of the time. Moreover, they are generally informative, capturing the true direction of causal effects in over half of all trials despite a relatively weak signal from the exposure $X$. Our bounds are clearly *correlated* with results from IV point estimators, but whereas competitors tend to overstate their confidence—bouncing between positive and negative causal effects multiple times in each panel—our bounds almost never stray so far as to miss the true ATE.

**Bayesian Methods.** An alternative family of methods for modeling latent parameters in the IV setting is based on Bayesian inference [Shapland et al., 2019, Bucur et al., 2020, Gkatzionis et al., 2021]. The goal in this approach is to estimate a posterior distribution for $\theta$, with partial identification bounds given by the upper and lower $\alpha$-quantiles of the credible interval. Rather than compare against some off the shelf method that does not explicitly encode our $\tau$-exclusion criterion, we design a Markov chain Monte Carlo (MCMC) sampler to model causal effects in the leaky IV setting (for details, see Appx. B.3). Due to the computa-

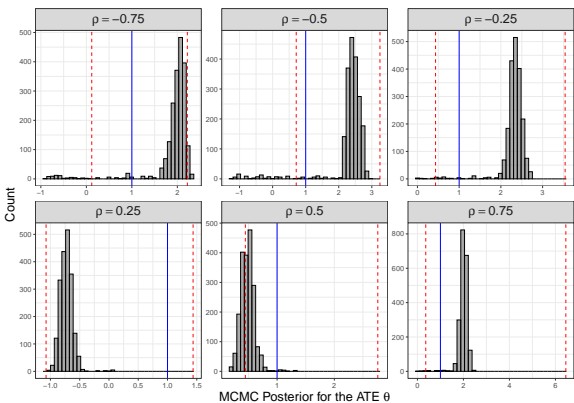

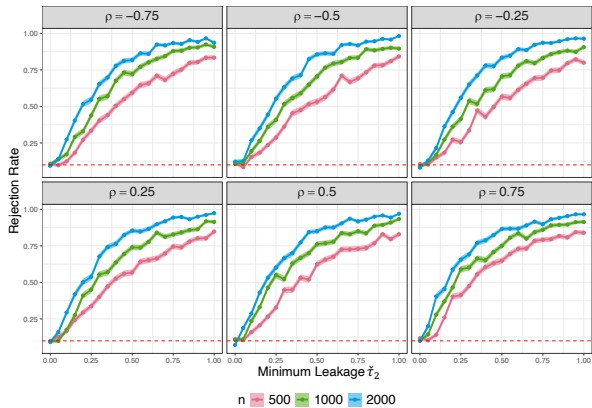

Figure 6: Comparison against a Bayesian model at a range of values for the confounding coefficient $\rho$. Histograms represent 2000 samples from a posterior distribution estimated via MCMC. The solid blue line denotes the true ATE, while the dashed red lines indicate LeakyIV bounds.

Figure 7: Power curves for the Monte Carlo exclusion test at varying values of the confounding coefficient $\rho$ and sample size $n$. Shading denotes standard errors. The horizontal dashed red line denotes the target level $\alpha = 0.1$.

tional demands of MCMC sampling, we focus on the case where $d_{\boldsymbol{Z}} = 5$, $\boldsymbol{\Sigma_{zz}}$ is diagonal, and the SNR for both $X$ and $Y$ is 2, varying only $\rho$ across a range of six possible values.

Results are presented in Fig. 6, featuring 2000 draws from the estimated posterior distribution for $\theta$. The blue line denotes the true ATE $\theta^* = 1$, while red dashed lines indicate bounds estimated by LeakyIV. We observe that even with uninformative priors on all linear parameters and just $n = 1000$ observations, the posterior tends to concentrate around a biased estimate, occasionally even placing some of the density outside our partial identification interval. Bayesian methods struggle in this setting because every solution in the feasible region has the same likelihood, which makes posteriors especially sensitive to the choice of prior distribution. Moreover, these methods do not decouple bounds on the causal parameter from the causal parameter itself. Any claims that posterior quantiles can be interpreted as "bounds" are either (i) a direct consequence of the prior; or (ii) heuristics that may be impossible to interpret outside the infinite data limit with an uninformative prior. Of course, this defeats the purpose of having priors on parameters in the first place.

**Power.** We run a series of power simulations to evaluate the sensitivity of our Monte Carlo exclusion test. With $d_{\boldsymbol{Z}} = 5$, $\theta^* = 1$, diagonal $\boldsymbol{\Sigma_{zz}}$, and fixed SNR = 2 for both $X$ and $Y$, we vary the sample size $n \in \{500, 1000, 2000\}$ and effect size $\check{\tau}_2 \in \{0, 0.1, \ldots, 1\}$ under six values of confounding $\rho$. We compute $p$-values using $B = 2000$ replicates and reject $H_0$ at level $\alpha = 0.1$. Empirical rejection rates are recorded over 500 runs (see Fig. 7). We find that type I error is controlled at the target level across all simulations, while power steadily increases with greater effect size, as expected. At $n = 2000$, we attain 95% power in all settings.

**Coverage.** Under the simulation settings of the Bayesian benchmark experiment, we evaluate nominal coverage using three bootstrap variants: the standard empirical distribution, a smoothed kernel estimate, and a Gaussian approximation. We generate 500 unique datasets for each setting and run 2000 bootstraps with fixed level $\alpha = 0.1$. Results are presented in Fig. 8. Empirical coverage is very close to the nominal 90% target in all settings, with a minimum of 0.886. The target level is always within a standard error of the mean across all trials. Though performance is similar for all three estimators, the Gaussian approximation appears slightly more conservative on average.

## 5 RELATED WORK

Violations of the exclusion restriction are well-documented in genetics [Hemani et al., 2018] and econometrics [Berkowitz et al., 2008]. One strategy for estimating causal effects in such settings is to permit a large number of potentially invalid instruments under the assumption that their average bias will tend to zero in the limit [Bowden et al., 2015, Kolesár et al., 2015]. With weak monotonicity constraints, these approaches can also provide nonparametric bounds on local average treatment effects [Flores and Flores-Lagunes, 2013].

Another family of methods starts from the assumption that some proportion of candidate instruments are valid and uses statistical procedures to focus on just those variables that satisfy (A1)-(A3). This can be achieved, for instance, via goodness of fit tests [Chu et al., 2001]; $L_1$-penalized regression for feature selection [Kang et al., 2016, Guo et al., 2018, Windmeijer et al., 2019]; independence tests for collider bias [Kang et al., 2020]; or modal validity assumptions in linear [Hartwig et al., 2017] and nonlinear [Hartford et al., 2021] IV models. Alternatively, data from multiple instruments can be pooled into a single variable using dimensionality

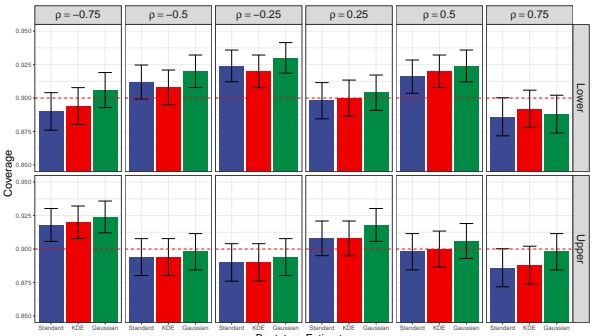

Figure 8: Empirical coverage of LeakyIV using three bootstrap estimators. Whiskers denote standard errors. The horizontal dashed red line denotes the nominal target of 90%.

reduction techniques [Kuang et al., 2020].

There is a substantial literature on Bayesian approaches to causal inference in IV settings. Lenkoski et al. [2014] use Bayesian model averaging to select IVs based on the strength of their association with the treatment, as codified by the relevance criterion (A1). Shapland et al. [2019] extend this method to account for linkage disequilibrium in Mendelian randomization experiments, which violate the no confounding condition (A2). More recently, several authors have proposed spike-and-slab priors to select genetic variants in the face of horizontal pleiotropy [Bucur et al., 2020, Gkatzionis et al., 2021], thereby addressing (A3).

Conley et al. [2012] propose ATE bounding methods given various kinds of prior information on leaky coefficients $\gamma$, including a range restriction and a (non-uniform) prior distribution. For the former, they return a union of confidence intervals resulting from a grid of possible values for $\gamma$. This method scales exponentially with $d_{\mathbf{Z}}$ and is potentially conservative as grid resolution grows finer. By contrast, our convex optimization approach is an efficient one-shot procedure that provides provably sharp bounds—in closed form, for the $L_2$ case. Their Bayesian proposal is similar to the method we compare against in Fig. 6.

The literature on tetrad constraints in linear SEMs goes back over a century [Spearman, 1904], although the method was revived and refined following the publication of Spirtes et al. [2000]'s tetrad representation theorem. Our exclusion test builds on generalized results developed by numerous authors [Shafer et al., 1995, Sullivant et al., 2010, Spirtes, 2013], although we are to our knowledge the first to propose a Monte Carlo inference procedure for testing such claims.

Partial identification intervals for counterfactual quantities can be computed exactly for discrete variables by formulating the problem as a polynomial program [Zhang et al., 2022, Duarte et al., 2023]. Though continuous data can always in principle be discretized with arbitrary precision, this quickly becomes intractable in the response function framework, as it leads to an exponential explosion of parameters.

Some continuous alternatives have been proposed with applications to IV models [Kilbertus et al., 2020, Hu et al., 2021, Padh et al., 2023]. However, the neural architectures underlying these models can be notoriously unstable, and are ill-suited to the linear SEM setting, which is standard in much biological and econometric research.

## 6 DISCUSSION

We have limited our analysis in this paper to linear SEMs. Though such linear models remain popular in many applications, it is well known that real-world systems often involve nonlinear dependencies between variables. To generalize the concept of leaky instruments, we could reformulate $\tau$-exclusion to place an upper bound on the conditional mutual information:

(A3'') *Generalized $\tau$-Exclusion:* $I(\mathbf{Z}; Y \mid X, \mathbf{U}) \leq \tau$.

Alternatively, (A3'') could place a bound on the gap between $p(y \mid x, \mathbf{u})$ and $p(y \mid x, \mathbf{u}, \mathbf{z})$ using some appropriate measure such as the Wasserstein distance or the KL-divergence. For binary $X, Y, Z$, Ramsahai [2012] and Silva and Evans [2016] represented this as a difference in expectations $|\mathbb{E}[Y \mid Z = 1, do(x)] - \mathbb{E}[Y \mid Z = 0, do(x)]| \leq \tau_z$. Extensions to vector-valued variants are conceptually straightforward. Estimating these quantities is more difficult than computing linear coefficients, but could help extend our approach to a wider class of data generating processes.

We have assumed in this paper that treatment effects are homogeneous throughout the population. However, a great deal of recent literature in causal machine learning has focused on *heterogeneous* treatment effects, where potential outcomes are presumed to vary as a function of pre-treatment covariates [Chernozhukov et al., 2018, Künzel et al., 2019, Nie and Wager, 2021]. Some authors have brought this framework into IV models, showing that tighter ATE bounds are possible with the help of observed confounders [Cai et al., 2007, Hartford et al., 2021, Levis et al., 2023]. Future work will consider *conditional* bounding methods, where extrema for $\theta$ may depend on instruments and/or other features causally antecedent to $X$.

A final note is that our method relies on user specification of the hyperparameter $\tau$. Ill-chosen thresholds may lead to issues, either in the form of overly conservative bounds (if $\tau$ is too high) or no bounds at all (if $\tau$ is below the minimum consistent with the data). In the worst case, invalid bounds will result from selecting a threshold that is high enough to satisfy the partial identification criterion but underestimates the true value of $\|\gamma\|_p$. However, we emphasize that $\tau$-exclusion is a strictly weaker assumption than the classical exclusion criterion (A3), which is widely applied—rightly or wrongly—in IV analyses. In many cases, setting $\tau = 0$—i.e., assuming perfectly exclusive instrumental variables—is

an essentially arbitrary choice, and a tempting one given the veneer of certainty provided by obtaining a point estimate for the ATE. If nothing else, we hope to make practitioners think twice before falling back on this familiar default. Background knowledge is regularly used to guide hyperparameter selection, and it is reasonable to assume that in many applications practitioners will have an *a priori* sense of how much information leakage is likely between $Z$ and $Y$. In particular, we can interpret this as a type of sensitivity analysis that asks how large $\tau$ can be such that the bounds exclude zero or effects of particular magnitude, and inquire with a practitioner whether larger values of $\tau$ are scientifically plausible.

# 7 CONCLUSION

We have presented a novel procedure for bounding causal effects in linear SEMs with unobserved confounding. By relaxing the exclusion criterion associated with the classical IV design, which often fails in many practical settings, our approach extends to a wide range of problems in genetic epidemiology, econometrics, and beyond. We introduce the notion of *leaky instruments*, which exert a limited direct effect on outcomes, and derive partial identifiability conditions for the ATE under minimal assumptions. Resulting bounds are sharp and practical, providing causal information in many cases where classical methods fail. We propose a Monte Carlo test that can falsify the exclusion criterion and a bootstrapping subroutine that guarantees asymptotic coverage at the target level. Future work will extend our results to multidimensional treatments, conditional bounding problems, and nonlinear systems, where alternative optimization strategies based on stochastic gradient descent may be required.

**Acknowledgments**

RS has been funded by the EPSRC Fellowship EP/W024330/1 and the EPSRC AI Hub EP/Y00759X/1.

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

# A PROOFS

## A.1 PROOF OF LEMMAS 1 AND 2

Consider Eqs. 1, 2, and 3, which define our model. Evaluating the product of $X$ and $\mathbf{Z}$ gives a relationship between covariances:

$$X\mathbf{Z} = \boldsymbol{\beta} \cdot \mathbf{Z}\mathbf{Z} + \epsilon_x \mathbf{Z}$$
$$\boldsymbol{\Sigma}_{xz} = \boldsymbol{\beta} \cdot \boldsymbol{\Sigma}_{zz}.$$

Solving for $\boldsymbol{\beta}$ gives

$$\boldsymbol{\beta} = \boldsymbol{\Sigma}_{xz} \cdot \boldsymbol{\Sigma}_{zz}^{-1}. \tag{4}$$

Likewise, the product of $X$ with itself gives

$$XX = \boldsymbol{\beta} \cdot \mathbf{Z}X + 2\boldsymbol{\beta} \cdot \mathbf{Z}\epsilon_x + \epsilon_x^2$$
$$\boldsymbol{\Sigma}_{xx} = \boldsymbol{\beta} \cdot \boldsymbol{\Sigma}_{zx} + \eta_x^2.$$

Using (4) and solving for $\eta_x^2$ gives

$$\eta_x^2 = \boldsymbol{\Sigma}_{xx} - \boldsymbol{\Sigma}_{xz} \cdot \boldsymbol{\Sigma}_{zz}^{-1} \cdot \boldsymbol{\Sigma}_{zx}$$
$$\eta_x^2 = \kappa_{xx}.$$

The product of $Y$ and $\mathbf{Z}$ gives

$$Y\mathbf{Z} = \boldsymbol{\gamma} \cdot \mathbf{Z}\mathbf{Z} + \theta X\mathbf{Z} + \epsilon_y \mathbf{Z}$$
$$\boldsymbol{\Sigma}_{yz} = \boldsymbol{\gamma} \cdot \boldsymbol{\Sigma}_{zz} + \theta\boldsymbol{\Sigma}_{xz}.$$

Solving for $\boldsymbol{\gamma}$ gives

$$\boldsymbol{\gamma} = \boldsymbol{\Sigma}_{zz}^{-1} \cdot \left(\boldsymbol{\Sigma}_{zy} - \theta\boldsymbol{\Sigma}_{zx}\right).$$

Taking the norm and using the definitions of $\boldsymbol{\alpha}, \boldsymbol{\beta}$, we recover Lemma 2:

$$\boxed{\|\boldsymbol{\gamma}\|_p = g_p(\theta) := \|\boldsymbol{\alpha} - \theta\boldsymbol{\beta}\|_p.} \tag{5}$$

The product of $Y$ and $X$ gives

$$YX = \theta XX + \boldsymbol{\gamma} \cdot \mathbf{Z}X + \epsilon_y X$$
$$\boldsymbol{\Sigma}_{yx} = \theta\boldsymbol{\Sigma}_{xx} + \boldsymbol{\gamma} \cdot \boldsymbol{\Sigma}_{zx} + \rho\eta_x\eta_y.$$

Using (5) and solving for $\rho\eta_x\eta_y$ gives

$$\rho\eta_x\eta_y = \boldsymbol{\Sigma}_{yx} - \theta\boldsymbol{\Sigma}_{xx} - \left(\boldsymbol{\Sigma}_{yz} - \theta\boldsymbol{\Sigma}_{xz}\right) \cdot \boldsymbol{\Sigma}_{zz}^{-1} \cdot \boldsymbol{\Sigma}_{zx}$$
$$\rho\eta_x\eta_y = \kappa_{xy} - \theta\kappa_{xx}. \tag{6}$$

Rearranging for $\eta_y^2$ gives

$$\eta_y^2 = \frac{\left(\kappa_{xy} - \theta\kappa_{xx}\right)^2}{\rho^2\eta_x^2}. \tag{7}$$

The product of $Y$ with itself gives

$$YY = \theta XY + \boldsymbol{\gamma} \cdot \boldsymbol{Z}Y + \epsilon_y Y$$

$$\Sigma_{yy} = \theta \Sigma_{xy} + \boldsymbol{\gamma} \cdot \Sigma_{zy} + \theta \rho \eta_x \eta_y + \eta_y^2.$$

Using (5), (6), and (7) gives

$$\Sigma_{yy} = \theta \Sigma_{xy} + \left( \Sigma_{yz} - \theta \Sigma_{xz} \right) \cdot \Sigma_{zz}^{-1} \cdot \Sigma_{zy}$$

$$+ \theta \left( \kappa_{xy} - \theta \kappa_{xx} \right) + \frac{\left( \kappa_{xy} - \theta \kappa_{xx} \right)^2}{\rho^2 \eta_x^2}.$$

Combining the $\Sigma$s into $\kappa$s,

$$\kappa_{yy} = \theta \kappa_{xy} + \theta \left( \kappa_{xy} - \theta \kappa_{xx} \right) + \frac{\left( \kappa_{xy} - \theta \kappa_{xx} \right)^2}{\rho^2 \eta_x^2},$$

collecting powers of $\theta$,

$$\kappa_{yy} - \frac{\kappa_{xy}^2}{\kappa_{xx} \rho^2} = 2\theta \kappa_{xy} \left( 1 - \frac{1}{\rho^2} \right) - \theta^2 \kappa_{xx} \left( 1 - \frac{1}{\rho^2} \right),$$

and massaging the $\rho$s and $\kappa_{xx}$s around, we have

$$\frac{\kappa_{yy} - \frac{\kappa_{xy}^2}{\kappa_{xx} \rho^2}}{1 - \frac{1}{\rho^2}} = 2\theta \kappa_{xy} - \theta^2 \kappa_{xx}$$

$$\frac{\rho^2 \kappa_{yy} \kappa_{xx} - \kappa_{xy}^2}{\rho^2 - 1} = 2\theta \kappa_{xx} \kappa_{xy} - \theta^2 \kappa_{xx}^2.$$

Solving this quadratic for $\theta \kappa_{xx}$ gives the solutions

$$\theta \kappa_{xx} = \kappa_{xy} \pm \sqrt{\kappa_{xy}^2 - \frac{\kappa_{xy}^2 - \rho^2 \kappa_{yy} \kappa_{xx}}{1 - \rho^2}}.$$

As $\rho > 0$ corresponds to the lower solution for $\theta$ (and vice versa), we have

$$\theta = \frac{1}{\kappa_{xx}} \left( \kappa_{xy} - \rho \sqrt{\frac{\kappa_{xx} \kappa_{yy} - \kappa_{xy}^2}{1 - \rho^2}} \right).$$

Exploiting the identity $\tan \left( \arcsin(x) \right) = x / \sqrt{1 - x^2}$ for $x \in [-1, 1]$, we derive the result stated in Lemma 1:

$$\boxed{\theta = f(\rho) = \kappa_{xx}^{-1} \left( \kappa_{xy} - \sqrt{\kappa_{xx} \kappa_{yy} - \kappa_{xy}^2} \tan \left( \arcsin(\rho) \right) \right).}$$

## A.2    PROOF OF LEMMA 3

We can think of the data covariance as constraining $\boldsymbol{\gamma}$ to lie in a 1-dimensional linear subspace $\boldsymbol{\alpha} - \theta \boldsymbol{\beta}$. (Recall that $\boldsymbol{\alpha}, \boldsymbol{\beta}$ are deterministic functions of $\Sigma$.) As $\boldsymbol{\alpha} \cdot \boldsymbol{\beta}$ may not equal zero in general, the resulting $L_p$ norm cannot be made arbitrarily small. Of course, computing the norm-minimizing coefficient is the definition of a linear regression task. Call the solution to this problem $\check{\theta}_p$ (where the index indicates optimization with respect to the $L_p$ norm). Since partial identification is only possible if information leakage exceeds the theoretical minimum consistent with the data covariance, we may define this lower bound as $\check{\tau}_p := g(\check{\theta}_p)$.

## A.3    PROOF OF LEMMA 4

Recall that $f$ gives $\theta$ as a function of $\rho$, while $g_p$ gives $\|\boldsymbol{\gamma}\|_p$ as a function of $\theta$. We define $h_p := g_p \circ f$ as a map from the confounding coefficient $\rho$ to the information leakage $\|\boldsymbol{\gamma}\|_p$. As $h_p$ is a continuous function with a compact domain, the extreme value theorem guarantees that a minimum exists. Moreover, since $f$ is bijective, we know by Lemma 3 that our target value $\check{\rho}_p$ must represent the inverse of $f$ evaluated at $\check{\theta}_p$. Setting $f$ to $\check{\theta}_p$ and solving for $\rho$, we derive the expression. We can now equivalently characterize the minimum leakage parameter as $\check{\tau}_p := h_p(\check{\rho}_p)$.

## A.4 PROOF OF THM. 1

Thm. 1 provides identifiability criteria for the leaky IV model. In the first part of the theorem, we describe a three-partition of the threshold space in terms of the theoretical leakage minimum $\check{\tau}_p$ and the oracle value $\tau_p^*$. We claim that the identifiability and validity of ATE bounds are fully characterized by where $\tau$ falls in relation to these parameters. Next, we show that point identification is possible if and only if latent parameters align in a specific way.

Take partial identifiability criteria first. The task of bounding the ATE in the leaky IV model amounts to finding the min/max values of $\theta$ that satisfy:

$$g_p(\theta) = \tau. \tag{8}$$

In other words, we fit a horizontal line $\|\boldsymbol{\gamma}\|_p = \tau$ across the function $g_p$ and report min/max points of intersection. Recall that $\check{\tau}_p$ is defined as the minimum of this function. Thus when $\tau$ falls below $\check{\tau}_p$, it is clear that there is no intersection between these two curves, and Eq. 8 has no solution. This is our sole partial identifiability criterion: provided all our structural assumptions hold, ATE bounds are well-defined if and only if $\tau \geq \check{\tau}_p$. Below this minimum leakage point lies *the infeasible region*.

But just because bounds are identifiable does not mean that they are valid. Suppose (for now) that the oracle threshold $\tau_p^*$ strictly exceeds the theoretical minimum, and recall the definition:

$$\tau_p^* := \|\boldsymbol{\gamma}^*\|_p = g_p(\theta^*),$$

where $\theta^*, \boldsymbol{\gamma}^*$ denote the true unobservable parameters. By the convexity of the norm, the solutions to Eq. 8 for any $\tau \in [\check{\tau}_p, \tau_p^*)$ will fail to capture the true ATE, as the resulting horizontal line lies below the point $(\|\boldsymbol{\gamma}^*\|_p, \theta^*)$. Even a $\boldsymbol{\Sigma}$-oracle—who, recall, has access to the population covariance matrix but *not* the latent parameters $\theta^*, \boldsymbol{\gamma}^*$—will return invalid bounds if queried with a threshold in this half-closed interval. For this reason, we call this band *the error region*.

With a threshold at or above the oracle value $\tau_p^*$, solutions to Eq. 8 are finally guaranteed to contain the true ATE $\theta^*$. This once again follows by convexity of $g_p$. Resulting bounds grow increasingly conservative with $\tau$. This *valid region* for all $\tau \geq \tau_p^*$ completes our three-partition of the threshold space.

We have thus far assumed that the oracle threshold *strictly* exceeds the theoretical minimum, but these parameters may coincide. If $\tau_p^* = \check{\tau}_p$, then the error region is empty and all identifiable bounds are valid. Moreover, if $g_p$ attains a unique minimum—as it must for all strictly convex norms, i.e. $p \in (1, \infty)$—then there exists just a single solution to Eq. 8 for $\tau = \tau_p^* = \check{\tau}_p$. In this case, lower and upper bounds for the ATE coincide and the causal parameter is point identified as $\theta^* = \check{\theta}_p$. Note that this fortuitous circumstance occurs with Lebesgue measure zero, as it imposes a nontrivial polynomial constraint on covariance parameters [Okamoto, 1973].

## A.5 PROOF OF COROLLARY 1.1

Under Eq. 2 and the exclusion criterion (A3), we must have that $\tau = \tau_p^* = 0$. Since $0 \leq \check{\tau}_p \leq \tau_p^*$, it follows that $\tau = \tau_p^* = \check{\tau}_p = 0$. This is a special case of the point identifiability result of Thm. 1. Define

$$\hat{X} := \mathbb{E}[X \mid \boldsymbol{Z}] = \boldsymbol{\beta} \cdot \boldsymbol{Z},$$

which represents the expected result of the first OLS regression. Then the 2SLS solution can be written as the ratio:

$$\theta^{2\text{SLS}} := \frac{\boldsymbol{\Sigma}_{\hat{x}y}}{\boldsymbol{\Sigma}_{\hat{x}\hat{x}}}$$

$$= \frac{\boldsymbol{\Sigma}_{zx} \cdot \boldsymbol{\Sigma}_{zy}}{\boldsymbol{\Sigma}_{zx} \cdot \boldsymbol{\Sigma}_{zx}}.$$

Exploiting the definitions $\boldsymbol{\alpha} := \boldsymbol{\Sigma}_{zz}^{-1} \cdot \boldsymbol{\Sigma}_{zy}$ and $\boldsymbol{\beta} := \boldsymbol{\Sigma}_{zz}^{-1} \cdot \boldsymbol{\Sigma}_{zx}$, we have:

$$\check{\theta}_2 = (\boldsymbol{\beta} \cdot \boldsymbol{\beta})^{-1} \boldsymbol{\beta} \cdot \boldsymbol{\alpha}$$

$$= \frac{(\boldsymbol{\Sigma}_{zz}^{-1} \cdot \boldsymbol{\Sigma}_{zx}) \cdot (\boldsymbol{\Sigma}_{zz}^{-1} \cdot \boldsymbol{\Sigma}_{zy})}{(\boldsymbol{\Sigma}_{zz}^{-1} \cdot \boldsymbol{\Sigma}_{zx}) \cdot (\boldsymbol{\Sigma}_{zz}^{-1} \cdot \boldsymbol{\Sigma}_{zx})}$$

$$= \frac{\boldsymbol{\Sigma}_{zx} \cdot \boldsymbol{\Sigma}_{zy}}{\boldsymbol{\Sigma}_{zx} \cdot \boldsymbol{\Sigma}_{zx}}$$

$$= \theta^{2\text{SLS}}.$$

Table 1: Contingency table of settings to consider for Thm. 2. Note that under non-strict convexity, an interval solution for $\check{\rho}_p$ is possible but not necessary; likewise, under non-strict leakage inequality, $\tau_p^* = \check{\tau}_p$ is possible but not necessary.

| Inequality | Convexity of $L_p$ norm | |
| | Strict | Non-strict |
| --- | --- | --- |
| Strict | Unique $\check{\rho}_p, \tau_p^* > \check{\tau}_p$ | Interval $\check{\rho}_p, \tau_p^* > \check{\tau}_p$ |
| Non-strict | Unique $\check{\rho}_p, \tau_p^* = \check{\tau}_p$ | Interval $\check{\rho}_p, \tau_p^* = \check{\tau}_p$ |

## A.6 PROOF OF THM. 2

We assume that partial identifiability criteria are met (see Thm. 1). Recall that $\theta$ is a bijective function of $\rho$ (see Lemma 1). To compute valid, sharp ATE bounds, it is therefore sufficient to show that there exist unique minimum and maximum values of $\rho$ such that $h_p(\rho) = \tau$. Call these $\rho_{\tau,p}^-$ and $\rho_{\tau,p}^+$, respectively. (Since the function $f$ is strictly decreasing, it maps the former to $\theta^+$ and the latter to $\theta^-$.) The advantage of working in $\rho$-space rather than $\theta$-space is that the confounding coefficient is guaranteed to lie on a compact interval that is independent of the data, namely $[-1, 1]$.

Recall that the $L_p$ norm is convex for all $p \geq 1$ and strictly convex for $p \in (1, \infty)$. Also, by definition, we have that $\tau_p^* \geq \check{\tau}_p$. Thus we have four possibilities to consider, with strict and non-strict variants of both convexity and the leakage inequality (see Table 1). Non-strict convexity raises complications due to the potential for plateaus in the $L_p$ norm; non-strict inequality raises complications if true and minimum leakage parameters coincide. We will show that $\rho_{\tau,p}^-$ and $\rho_{\tau,p}^+$ are uniquely identified in all four settings.

Start with the simplest case, in which both the convexity and inequality are strict. In this setting, we have exactly two solutions to the equation $h_p(\rho) = \tau$, one on either side of $\check{\rho}_p$, which is the unique minimizer of $h_p$. Thus one solution lies on the interval $[-1, \check{\rho}_p]$, and another on $[\check{\rho}_p, 1]$.[3] This establishes the existence and uniqueness of $\rho_{\tau,p}^-$ and $\rho_{\tau,p}^+$.

Now consider the case where $h_p$ is strictly convex but the true leakage coincides with the theoretical minimum (lower left quadrant of Table 1). In this case, we have just a single solution to the equation $h_p(\rho) = \tau$, namely $\check{\rho}_p$. This implies that $\check{\rho}_p = \rho_{\tau,p}^- = \rho_{\tau,p}^+$.

Greater care is required when $h_p$ is not strictly convex, as we can no longer assume the uniqueness of $\check{\rho}_p$ or that $h_p(\rho) = \tau$ has at most two solutions. However, when no unique minimum exists for a convex function with a compact domain, the set of minimizing solutions forms a compact interval. (This follows from the extreme value theorem.) Consider the setting where the leakage inequality is strict but no single value of $\rho$ minimizes $h_p$ (upper right quadrant of Table 1). We can select any value from the compact interval $\check{\rho}_p$ and use this to partition $[-1, 1]$, since strict inequality guarantees that any solution must intersect with $h_p$ above its minimum. Still, we may have have uncountably many solutions to the equation $h_p(\rho) = \tau$ if $\tau$ aligns with a plateau in the norm on one or both sides of $\check{\rho}_p$. Convexity guarantees that we will have at most two sets of solutions, one on either side of the minimum. Call these intervals $\rho_0$ and $\rho_1$. Since both are closed, each contains a unique min/max. Our target parameters are therefore identified by taking the extreme values of each, i.e. setting $\rho_{\tau,p}^- = \min \rho_0$ and $\rho_{\tau,p}^+ = \max \rho_1$.

Finally, consider the case where neither the convexity of the $L_p$ norm nor the leakage inequality is strict (lower right quadrant of Table 1). This is arguably simpler than the setting with strict inequality and non-strict convexity, since we have just a single compact interval of solutions at $\check{\rho}_p$. Our target parameters in this case are identified via $\rho_{\tau,p}^- = \min \check{\rho}_p$ and $\rho_{\tau,p}^+ = \max \check{\rho}_p$.

## A.7 PROOF OF COROLLARY 2.1

To find ATE bounds with an $L_2$ threshold on information leakage, we invoke Lemma 2 and find that leakage is quadratic in $\theta$:

$$\|\boldsymbol{\gamma}\|_2^2 = \|\boldsymbol{\beta}\|_2^2\, \theta^2 - 2\boldsymbol{\alpha} \cdot \boldsymbol{\beta}\, \theta + \|\boldsymbol{\alpha}\|_2^2.$$

---

[3]In fact, when $\tau_p^* > \check{\tau}_p$, we know that $\check{\rho}_p$ is not a viable solution, and so we can replace the closed intervals with half-open intervals $[-1, \check{\rho}_p)$ and $(\check{\rho}_p, 1]$. Since this is not the case when $\tau_p^* = \check{\tau}_p$, we stick with closed intervals throughout for greater generality.

We set $\tau = \|\boldsymbol{\gamma}\|_2$ and solve for $\theta$ using the quadratic formula:

$$\theta = \frac{2\boldsymbol{\alpha} \cdot \boldsymbol{\beta} \pm \sqrt{(2\boldsymbol{\alpha} \cdot \boldsymbol{\beta})^2 - 4\|\boldsymbol{\beta}\|_2^2 \left(\|\boldsymbol{\alpha}\|_2^2 - \tau^2\right)}}{2\|\boldsymbol{\beta}\|_2^2}.$$

Observe that the first summand reduces to the norm-minimizing ATE value identified in Lemma 3:

$$\frac{2\boldsymbol{\alpha} \cdot \boldsymbol{\beta}}{2\|\boldsymbol{\beta}\|_2^2} = (\boldsymbol{\beta} \cdot \boldsymbol{\beta})^{-1} \boldsymbol{\beta} \cdot \boldsymbol{\alpha} =: \check{\theta}_2.$$

Some light simplifications and rearrangements renders the final expression:

$$\boxed{\check{\theta}_2 \pm (\boldsymbol{\beta} \cdot \boldsymbol{\beta})^{-1} \sqrt{(\boldsymbol{\beta} \cdot \boldsymbol{\beta})(\tau^2 - \boldsymbol{\alpha} \cdot \boldsymbol{\alpha}) + (\boldsymbol{\alpha} \cdot \boldsymbol{\beta})^2}}$$

## A.8  PROOF OF THM. 3

Let $\mathcal{M}$ be the space of all models satisfying our structural constraints—Eqs. 1, 2, 3 and assumptions (A1), (A2), and (A3$'_s$)—for some fixed distribution family $\mathcal{P}$ and $d_{\boldsymbol{Z}} \geq 2$. (Recall that (A3$'_s$) is consistent with the classic exclusion criterion (A3) under $\tau = 0$.) We partition $\mathcal{M}$ into null and alternative classes $\mathcal{M}_0, \mathcal{M}_1$ depending on whether the models in each satisfy $H_0 : \psi = 0$. We reiterate that this condition is necessary but not sufficient to guarantee (A3). Each dataset $\mathcal{D}_n$ is sampled from some fixed but unknown $P_{\boldsymbol{\Sigma}}$ that belongs to either $\mathcal{M}_0$ or $\mathcal{M}_1$.

For every $P_{\boldsymbol{\Sigma}} \in \mathcal{M}$, there exists some nearest null neighbor $Q_{\boldsymbol{\Sigma}}^* \in \mathcal{M}_0$ (not necessarily unique) satisfying

$$Q_{\boldsymbol{\Sigma}}^* := \underset{Q_{\boldsymbol{\Sigma}} \in \mathcal{M}_0}{\arg\min} D_{KL}(P_{\boldsymbol{\Sigma}} \| Q_{\boldsymbol{\Sigma}}).$$

Of course, when $P_{\boldsymbol{\Sigma}} \in \mathcal{M}_0$, we have $P_{\boldsymbol{\Sigma}} = Q_{\boldsymbol{\Sigma}}^*$ and the KL-divergence goes to zero. Let $f_\psi : \mathbb{R}^{n \times (2+d_{\boldsymbol{Z}})} \mapsto \mathbb{R}_{\geq 0}$ be a function from input data to corresponding test statistics $\psi$. (The bounded range follows from the fact that all entries in the matrix $\Lambda \cdot \Lambda$ are non-negative.) Let $\mathcal{D}_n$ be a dataset sampled from $P_{\boldsymbol{\Sigma}}$, and let $G_n^{\boldsymbol{\Sigma}}$ be the sampling distribution of $\psi$ at sample size $n$, i.e. $\hat{\psi}_n = f_\psi(\mathcal{D}_n) \sim G_n^{\boldsymbol{\Sigma}}$ for any $\mathcal{D}_n \sim P_{\boldsymbol{\Sigma}}$. We denote the corresponding null distribution as $G_n^{\boldsymbol{\Sigma}^0}$, which represents the sampling distribution of $\psi$ under $H_0$ at sample size $n$, i.e. $\psi_n^0 = f_\psi(\mathcal{D}_n^0) \sim G_n^{\boldsymbol{\Sigma}^0}$, for null datasets $\mathcal{D}_n^0 \sim Q_{\boldsymbol{\Sigma}}^*$.

To establish that $p_{\text{MC}}$ is an asymptotically valid $p$-value against $H_0$, it suffices to show that the Monte Carlo null distribution $\hat{G}_n^{\boldsymbol{\Sigma}^0}$ converges to $G_n^{\boldsymbol{\Sigma}^0}$. This follows from the validity of our procedure for constructing the null covariance matrix $\boldsymbol{\Sigma}^0$, which involves the minimum perturbation required to guarantee $H_0$. Specifically, we impose a linear dependence between covariance vectors $\boldsymbol{\Sigma}_{\boldsymbol{zx}}$ and $\boldsymbol{\Sigma}_{\boldsymbol{zy}}$ using the scaling factor $\hat{\theta}^{\text{2SLS}}$. Thus $\boldsymbol{\Sigma}^0$ satisfies $\psi = 0$ by construction. Moreover, since this is achieved by changing as few parameters as possible by as little as possible, there exists no nearer neighbor to $P_{\boldsymbol{\Sigma}}$ within $\mathcal{M}_0$ than the resulting distribution, which therefore satisfies our definition of $Q_{\boldsymbol{\Sigma}}^*$.

We assume access to some method for sampling from $Q_{\boldsymbol{\Sigma}}^*$, e.g. via $\mathcal{N}(\mathbf{0}, \boldsymbol{\Sigma}^0)$ if $\mathcal{P}$ is the family of mean-zero multivariate Gaussians. We draw $B$ many datasets of size $n$ from $Q_{\boldsymbol{\Sigma}}^*$ and record resulting test statistics to generate the synthetic null distribution $\hat{G}_n^{\boldsymbol{\Sigma}_0}$. Convergence is assured when $p_{\text{MC}}$ is uniformly distributed under $H_0$. Let $c_\alpha(\mathcal{D}_n)$ denote the critical value at type I error rate $\alpha$ for dataset $\mathcal{D}_n$, such that, under $H_0$, the rejection region of statistics

$$R_\alpha(\mathcal{D}_n) = \left\{\psi_n : \psi_n \geq c_\alpha(\mathcal{D}_n)\right\}$$

integrates to $\alpha$. Rejection regions are nested for our one-sided test, i.e. $R_\alpha \subset R_{\alpha'}$ if $\alpha < \alpha'$. Thus $c_\alpha(\mathcal{D}_n)$ represents the $1 - \alpha$ quantile of the null distribution $G_n^{\boldsymbol{\Sigma}_0}$. We reject $H_0$ if $p_{\text{MC}} \leq \alpha$, resulting in the identity:

$$p_{\text{MC}} = p_{\text{MC}}(\mathcal{D}_n) = \inf\left\{\alpha : \psi_n \in R_\alpha(\mathcal{D}_n)\right\}.$$

Then for all $\alpha \in (0, 1)$, we have

$$\mathbb{P}_{\mathcal{D}_n \sim Q_{\boldsymbol{\Sigma}}^*}\left(\psi_n \in R_\alpha(\mathcal{D}_n)\right) = \alpha,$$

which implies that:

$$\mathbb{P}_{\mathcal{D}_n \sim Q_{\boldsymbol{\Sigma}}^*}\left(p_{\text{MC}}(\mathcal{D}_n) \leq u\right) = u$$

for all $u \in [0, 1]$ [Lehmann and Romano, 2005]. In other words, $p_{\text{MC}}$ is uniformly distributed under $H_0$, as desired, and the Monte Carlo null distribution has converged on the target $G_n^{\boldsymbol{\Sigma}^0}$. We add one to the numerator and denominator as a necessary finite sample adjustment.

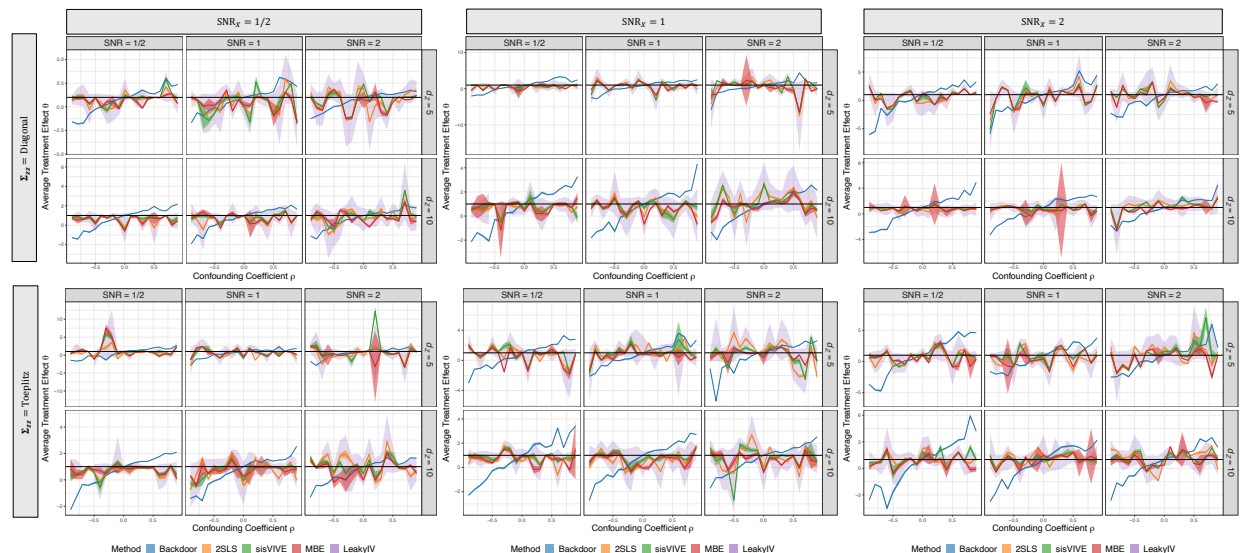

Figure 9: Complete results for the benchmark experiment against point estimators in Sect. 4. The top two rows uses a diagonal covariance matrix; the bottom two use a Toeplitz covariance matrix. For each value of $\text{SNR}_X$, we consider three unique values of $\text{SNR}_Y$ (simply labelled SNR within the facet grid).

## A.9 PROOF OF THM. 4

It may not be immediately clear that bootstrapping is appropriate for ATE bounds. After all, it is well known that the bootstrap cannot provide a valid sampling distribution for fixed order statistics such as ranks, or a target parameter that lies on the boundary of the parameter space [Andrews, 2000]. But though we refer to our bounds as "min" and "max" solutions, that does not mean they are calculated via fixed order statistics. On the contrary, each represents a continuous solution to a differentiable optimization task, not the smallest or largest element in a discrete set. The only boundary condition for our target parameters is $\theta^- \leq \theta^+$, which automatically holds under the partial identifiability criterion $\tau \geq \check{\tau}_p$. As our estimator is undefined for samples that violate the criterion, our sampling distribution is always conditioned on this event.

In general, any statistic that is a differentiable function of sample moments admits an Edgeworth expansion and can therefore have its distribution consistently estimated via bootstrap resampling [Hall, 1992, Davison and Hinkley, 1997]. Recall that our ATE bounds represent the intersection of (a) a $\tau$-feasible region that is fixed *a priori*; and (b) the $L_p$ norm of $\gamma = \alpha - \theta\beta$, where the latter two vectors are defined as dot products of covariance parameters. Resulting bounds vary smoothly under resampling, since $\alpha$ and $\beta$ are differentiable with respect to $\Sigma$. Though more generalized relaxations of the exclusion criterion may introduce discontinuities or other issues, our formulation of $\tau$-exclusion poses no such difficulties. The resulting bootstrap distributions are asymptotically valid and practically useful, providing statistical inference without any parametric assumptions.

Of course, it is perfectly possible that some bootstrap samples may have to be discarded if the intersection of regions (a) and (b) is empty. In such cases, we simply restrict attention to those bootstraps that satisfy the partial identifiability criterion $\tau > \check{\tau}_p$, which should represent a non-negligible proportion of all bootstraps if the inequality is satisfied in the original dataset. This procedure is akin to sampling under a feasibility condition, and requires no extra steps to maintain bootstrap consistency, as in Andrews [2000] or Ramsahai and Lauritzen [2011].

## B  EXPERIMENTS

### B.1  BENCHMARKS

We implement the backdoor adjustment via simple linear regression. Similarly, the 2SLS estimator is computed using OLS. We use the CRAN implementation of sisVIVE. R code for MBE was provided by the authors. Results of benchmark experiments for all simulation configurations are presented in Fig. 9.

## B.2 INTERPRETING THE SCALES OF STRUCTURAL PARAMETERS

In our experiments, we fix the true causal effect $\theta^* = 1$ and tune the signal-to-noise ratios (SNRs) for $X$ and $Y$, denoted here as $\text{SNR}_X$ and $\text{SNR}_Y$ respectively. We show that these quantities, coupled with the variance $\boldsymbol{\Sigma}_{yy}$, the magnitude of $\boldsymbol{\beta}$ and the instruments covariance matrix $\boldsymbol{\Sigma}_{zz}$, uniquely define the magnitude of the remaining structural parameters $(\boldsymbol{\gamma}, \boldsymbol{\Sigma}_{yy}, \eta_x, \eta_y)$ at a given level of confounding $\rho$, when the directions $(\boldsymbol{\gamma}, \boldsymbol{\beta})$ are randomized. The choice to tune the dimensionless parameters $\text{SNR}_X$ and $\text{SNR}_Y$ provides a more interpretable grid search than we would have if we were to vary the structural parameters directly. This is especially true given that the directions of the vector-valued structural parameters—randomized in our experiments and exponentially hard to search through in a high-$d_Z$ setting—play a crucial role in the effect in the proportions of each observable's variance explained by each causal effect.

In our setting, $\boldsymbol{\beta}$ and $\boldsymbol{\gamma}$ are randomized through $\boldsymbol{\beta} = \tilde{\boldsymbol{\beta}}$ and $\boldsymbol{\gamma} = \zeta\tilde{\boldsymbol{\gamma}}$, where the components $\tilde{\beta}_i$ and $\tilde{\gamma}_i$, $i \in [d_Z]$ are drawn identically and independently (iid) from the the standard normal distribution. We show that $\eta_x, \eta_y, \zeta, \boldsymbol{\Sigma}_{xx}$ are uniquely determined through the equations for the variances,

$$\boldsymbol{\Sigma}_{xx} = \tilde{\boldsymbol{\beta}} \cdot \boldsymbol{\Sigma}_{zz} \cdot \tilde{\boldsymbol{\beta}} + \eta_x^2,$$
$$\boldsymbol{\Sigma}_{yy} = \zeta^2 \tilde{\boldsymbol{\gamma}} \cdot \boldsymbol{\Sigma}_{zz} \cdot \tilde{\boldsymbol{\gamma}} + \theta^2 \boldsymbol{\Sigma}_{xx} + 2\theta\zeta\tilde{\boldsymbol{\gamma}} \cdot \boldsymbol{\Sigma}_{zz} \cdot \tilde{\boldsymbol{\beta}} + 2\theta\eta_x\eta_y\rho + \eta_y^2,$$

and the signal-to-noise ratios,

$$\text{SNR}_X := \frac{\boldsymbol{\beta} \cdot \boldsymbol{\Sigma}_{zz} \cdot \boldsymbol{\beta}}{\eta_x^2} = \frac{1}{\eta_x^2}\tilde{\boldsymbol{\beta}} \cdot \boldsymbol{\Sigma}_{zz} \cdot \tilde{\boldsymbol{\beta}},$$
$$\text{SNR}_Y := \frac{\boldsymbol{\gamma} \cdot \boldsymbol{\Sigma}_{zz} \cdot \boldsymbol{\gamma} + \theta^2\boldsymbol{\Sigma}_{xx} + 2\theta\boldsymbol{\gamma} \cdot \boldsymbol{\Sigma}_{zz} \cdot \boldsymbol{\beta}}{2\theta\eta_x\eta_y\rho + \eta_y^2} = \frac{\boldsymbol{\Sigma}_{yy}}{\eta_y^2 + 2\theta\rho\eta_x\eta_y} - 1.$$

Note that "noise" is any contribution involving the unobserved confounding $\boldsymbol{\Sigma}_{\epsilon\epsilon}$. In a certain sense, the definition of $\text{SNR}_x$ is ambiguous in our setting because $\eta_x^2 = \kappa_{xx}$ can be determined from generated data. We stress our choice of the definition here: if we took $\eta_x^2$ to be "signal" then $\text{SNR}_X$ would be infinite.

We solve these four coupled quadratic equations for the remaining parameters $\eta_x, \eta_y, \zeta, \boldsymbol{\Sigma}_{xx}$. They have unique solutions if we demand each of these scaling factor and the standard deviations $\eta_x, \eta_y$ to be positive. We choose to write the solutions to these equations in the following form:

$$\eta_x = \sqrt{\frac{A_{xx}}{\text{SNR}_X}}, \tag{9}$$

$$\boldsymbol{\Sigma}_{xx} = \frac{\text{SNR}_X + 1}{\text{SNR}_X}, \tag{10}$$

$$\eta_y = \theta\rho\eta_x\left(-1 + \sqrt{1 + \frac{\boldsymbol{\Sigma}_{yy}}{1 + \text{SNR}_Y}}\right), \tag{11}$$

$$\zeta = \frac{\theta A_{yy}}{A_{xy}}\left(-1 + \sqrt{1 + \frac{A_{xy}}{A_{yy}^2\theta^2}\left(\frac{\boldsymbol{\Sigma}_{yy}}{1 + \frac{1}{\text{SNR}_Y}} - \theta^2\boldsymbol{\Sigma}_{xx}\right)}\right), \tag{12}$$

where

$$A_{xx} := \tilde{\boldsymbol{\beta}} \cdot \boldsymbol{\Sigma}_{zz} \cdot \tilde{\boldsymbol{\beta}},$$
$$A_{xy} := \tilde{\boldsymbol{\beta}} \cdot \boldsymbol{\Sigma}_{zz} \cdot \tilde{\boldsymbol{\gamma}},$$
$$A_{yy} := \tilde{\boldsymbol{\gamma}} \cdot \boldsymbol{\Sigma}_{zz} \cdot \tilde{\boldsymbol{\gamma}}.$$

Notice that the term in the brackets in the equation for $\zeta$ is the signal due to $\tau$-exclusion, i.e., the residual signal not solely due to $\theta$. This term is, therefore, always greater than 1, so $\zeta$ is always greater than 0. Notice also that these equations are more general than in our particular experimental setting since we have left $\theta = \theta^*$, $\boldsymbol{\Sigma}_{zz}$ and $\boldsymbol{\Sigma}_{yy}$ to be decided.

Inputting the above solutions, in order, to a data generating process allows us to tune these terms by specifying the signal-to-noise ratios. As a final note, one may be interested in studying asymptotic regimes in which the variance of $X$ or $Y$ is dominated either by signal or noise. These equations, or equations very similar to these, allow for the rigorous study of linear models in such regimes through asymptotic expansions with respect to the SNRs.

## B.3 BAYESIAN BASELINE

One of our choices of a method to compare against ours is a full-likelihood Gaussian model with Bayesian posteriors with a bounded L2 norm on $\gamma$.

Assume all variables follow a joint multivariate zero-mean Gaussian distribution. Recall that that $\boldsymbol{Z}$ are candidate instruments, $X$ is the treatment, $Y$ is the outcome.

Let $\boldsymbol{\gamma}$ be the coefficients of $\boldsymbol{Z}$ in the equation for $Y$, and $\tau$ is such that $||\boldsymbol{\gamma}||_2 \leq \tau$. We will encode $\boldsymbol{\gamma}$ as

$$\boldsymbol{\gamma} := \boldsymbol{b} \times \sqrt{\frac{\kappa \times \tau}{||\boldsymbol{b}||_2^2}},$$

where $\kappa \in [0, 1]$ is another (redundant) parameter, and $\boldsymbol{b}$ is the free parameter vector of the same dimensionality as $\boldsymbol{\gamma}$. The interpretation is that $\kappa \times \tau$ is the norm of $\boldsymbol{\gamma}$, and $\boldsymbol{b}$ is a direction vector. This provides a direct comparison against our constrained optimization method, as both methods are capable of directly using information about the norm of $\boldsymbol{\gamma}$ and we will below put an uniform prior on $\kappa$.

Assuming $\boldsymbol{Z}$ below is a row vector, the model is:

$$
\begin{aligned}
\boldsymbol{Z} &\sim MVN(0, \boldsymbol{\Sigma_{zz}}) \\
X &= \boldsymbol{\beta} \cdot \boldsymbol{Z} + \epsilon_x \\
Y &= \boldsymbol{\gamma} \cdot \boldsymbol{Z}\theta X + \epsilon_y \\
(\epsilon_x, \epsilon_y) &\sim MVN(0, \boldsymbol{\Sigma_{\epsilon\epsilon}}),
\end{aligned}
$$

where $\boldsymbol{\Sigma_{zz}}$ and $\boldsymbol{\Sigma_{\epsilon\epsilon}}$ are generic positive definite matrices, and $MVN$ means multivariate Gaussian distribution. The parameter set $\Theta$ is $\{\boldsymbol{\beta}, \kappa, \boldsymbol{b}, \theta, \boldsymbol{\Sigma_{zz}}, \boldsymbol{\Sigma_{\epsilon\epsilon}}\}$.

In what follows, we will consider only the independent candidate instruments case

$$
\boldsymbol{\Sigma_{zz}} := \begin{bmatrix} \eta_{z1}^2 & 0 & 0 & \ldots & 0 \\ 0 & \eta_{z2}^2 & 0 & \ldots & 0 \\ 0 & 0 & 0 & \ldots & \eta_{z_{d_Z}}^2 \end{bmatrix},
$$

and parameterize $\boldsymbol{\Sigma_{\epsilon\epsilon}}$ as

$$
\boldsymbol{\Sigma_{\epsilon\epsilon}} := \begin{bmatrix} \eta_x^2 & \rho\eta_x\eta_y \\ \rho\eta_x\eta_y & \eta_y^2 \end{bmatrix},
$$

for $\rho \in [-1, 1]$. The independence assumption on $\boldsymbol{Z}$ is merely to simplify our sampler code.

Priors are defined as follows:

$$
\begin{aligned}
\kappa &\sim U(0, 1) \\
\boldsymbol{\beta} &\sim MVN(0, I \times v_\beta) \\
\boldsymbol{b} &\sim MVN(0, I \times v_b) \\
\theta &\sim N(0, v_\theta) \\
\eta_{zi}^2 &\sim logN(l_{\mu_z}, l_{v_z}), \text{for } i = 1, 2, \ldots, n_Z \\
\eta_x^2 &\sim logN(l_{\mu_x}, l_{v_x}) \\
\eta_y^2 &\sim logN(l_{\mu_y}, l_{v_y}) \\
\rho &\sim U(-1, 1)
\end{aligned}
$$

where $I$ is the identity matrix of corresponding dimensionality, $U$ is the uniform distribution on the unit interval, and $logN$ denotes the log-normal distribution. Remaining symbols $v_\beta, v_b, v_\theta, l_{\mu_z}, l_{v_z}, l_{\mu_x}, l_{v_x}, l_{\mu_y}$, and $l_{v_y}$ are hyperparameters.

Given a dataset $D$ with each of its $n$ rows denoting a data point, the sufficient statistic for this model is

$$S := D \cdot D.$$

The *model covariance matrix* $\boldsymbol{\Sigma}(\Theta)$ is given by

$$
\begin{aligned}
\boldsymbol{\Sigma}(\Theta)_{\boldsymbol{zz}} &:= \boldsymbol{\Sigma}_{\boldsymbol{zz}} \\
\boldsymbol{\Sigma}(\Theta)_{\boldsymbol{zx}} &:= \boldsymbol{\Sigma}_{\boldsymbol{zz}}\beta \\
\boldsymbol{\Sigma}(\Theta)_{\boldsymbol{xx}} &:= \boldsymbol{\beta} \cdot \boldsymbol{\Sigma}_{\boldsymbol{zz}} \cdot \beta + \eta_x^2 \\
\boldsymbol{\Sigma}(\Theta)_{\boldsymbol{xy}} &:= \Sigma(\Theta)_{\boldsymbol{zx}} \cdot \boldsymbol{\gamma} + \eta_x^2 \times \theta + \eta_{xy} \\
\boldsymbol{\Sigma}(\Theta)_{\boldsymbol{zy}} &:= \boldsymbol{\Sigma}_{\boldsymbol{zz}} \cdot \boldsymbol{\gamma} + \boldsymbol{\Sigma}(\Theta)_{\boldsymbol{zx}} \times \theta \\
\boldsymbol{\Sigma}(\Theta)_{\boldsymbol{yy}} &:= \boldsymbol{\gamma} \cdot \boldsymbol{\Sigma}_{\boldsymbol{zz}} \cdot \boldsymbol{\gamma} + 2 \times \boldsymbol{\gamma} \cdot \Sigma_{\boldsymbol{zx}}(\Theta) \times \theta + \\
&\qquad \theta^2 \times \Sigma(\Theta)_{xx} + 2 \times \theta \times n_{xy} + \eta_y^2.
\end{aligned}
$$

Given that, the log-likelihood function is

$$
L(\Theta) := -0.5 \times trace(\boldsymbol{\Sigma}(\Theta)^{-1}S) - 0.5 \times n \times \log(|\boldsymbol{\Sigma}(\Theta)|),
$$

where the columns/rows of $S$ are sorted in the same way as the columns/rows of $\Sigma(\Theta)$. We use a plain random walk Metropolis-Hastings method to sample from the posterior of this distribution. We implement the uniform priors by the encoding $\kappa := \Phi(W_\kappa)$, where $\Phi(\cdot)$ is the standard Gaussian cdf and $W_\kappa$ is a standard Gaussian random variable to be sampled. Likewise, $\rho := 2 \times \Phi(W_\rho) - 1$.

### B.3.1 Usage

Like MASSIVE [Bucur et al., 2020], this is a full Bayesian approach that returns a full posterior on $\theta$. Unlike MASSIVE, which is designed for soft sparsity constraints, our prior on $\gamma$ is on a Gaussian distributed disc with the radius being given a uniform prior on $[0, \tau]$ so that it is the closer match to the principle behind our hard constrained optimization method.

For a comparison to take place, we translate the posterior distribution over $\theta$ as "bounds". More precisely, let $q_{\theta_\alpha}$ be the $\alpha$-th quantile of the posterior distribution of $\theta$. A "lower bound" here is taken to mean the $[-\infty, q_{\theta_\alpha}]$ interval, although it is not explicitly made to "capture" the population lower bound with probability $\alpha$. That can only happen in a heuristic sense, as the full Bayesian approach is oblivious to non-identifiability issues and has no sense what a "population lower bound" should be. For joint "capturing" of lower and upper bounds, we suggest $[-\infty, q_{\theta_{\alpha/2}}]$ along with $[q_{\theta_{1-\alpha/2}}, +\infty]$.

Like MASSIVE, the posterior on $\theta$ never converges to a single point in the limit of the infinite data, and any entropy left in that case is a consequence of the prior. If the prior is informative, it is entirely possible for the posterior to exclude the true $\theta$ even if it perfectly fits the population distribution. If the prior is uninformative, the limiting posterior support should coincide with the feasible region obtained by plugging in the population covariance matrix, although any curvature of this posterior within its support is an artefact of the prior. However, with finite data, there is no clear way of separating entropy due to probabilistic uncertainty from entropy due to unidentifiability. Any claims that tails of this distribution have a correspondence to partial identifiability results is an ill-posed heuristic at best.