# OpenReview forum: "Bounding causal effects with leaky instruments"
_auai.org/UAI/2024/Conference — UAI 2024 poster_

### Official Review · Reviewer_HFN4 · 2024-03-11

**Q2-1 Originality-Novelty:** 3
**Q2-2 Correctness-Technical Quality:** 3
**Q2-5 Clarity Of Writing:** 3

**Q1 Summary And Contributions:**

This paper presents a method to bound the ATE in the presence of an approximate instrumental variable, named leaky instrument.

**Q2-3 Extent To Which Claims Are Supported By Evidence:**

2: Fair: the main claims are somewhat supported by evidence (but the experimental evaluation may be weak, or does not match entirely with the claims, important baselines may be missing, proofs contain important ideas but lack rigor, algorithmic details are only discussed superficially, references are imprecise, assumptions are not sufficiently motivated or explicated, etc.).

**Q2-4 Reproducibility:**

3: Good: key resources (e.g. proofs, code, data) are available and key details (e.g. proofs, experimental setup) are sufficiently well-described for competent researchers to confidently reproduce the main results.

**Q3 Main Strengths:**

The problem addressed is important, namely bounding the ATE under non-ideal conditions such as the violation of some of the assumptions behind instrumental variables.

**Q4 Main Weakness:**

There is some kind of contradiction in relaxing the assumptions behind instrumental variables by allowing leakage as this work does, and assuming a linear model as this work does too. In other words, while the authors deserve for the theoretical results presented, I believe that their practical relevance is limited since a linear model assumed is typically a too crude model of reality.

**Q5 Detailed Comments To The Authors:**

The authors do mention in the conclusions that future work may include extending their work to nonlinear models. I would have appreciated some elaboration on this, since I do not think it is trivial.

**Q9 Complying With Reviewing Instructions:**

Yes

---

> ### Author Rebuttal · Authors · 2024-04-02
>
> We thank HFN4 (henceforth R5) for their thoughtful review. R5’s main objection appears to be with our restriction to linear structural equation models (SEMs). While we appreciate this concern, we do explicitly discuss this limitation in our introduction, and again in the discussion section (where we also discuss a route towards relaxing this assumption).  We would also like to emphasize that decades of prominent research in the social and natural sciences relies on linear regression. For instance, this is the foundation of the two stage least squares model, which is widely deployed in econometrics and genetic epidemiology. On the methodological side, we observe that much of the foundational work in causal discovery and inference was founded on the study of linear SEMs (Spirtes et al., 2000).
>
> We do not intend to suggest that linear SEMs are in any sense universal or optimal. However, we make two claims: (1) Many real-world systems are well described by linear SEMs. This is attested by decades of applied research. For instance, the linear IV model has been successfully used to study the causal effect of various interventions on downstream wages, such as military conscription (Angrist, 1990), job training (Abadie et al., 2003), and compulsory schooling (Angrist & Imbens, 1995). The 2021 Nobel Prize in Economic Sciences was awarded to Joshua Angrist and Guido Imbens, due in no small part to their groundbreaking work on causal inference in linear SEMs. (2) Getting matters right in the linear setting is a fundamental step toward generalizing to the nonlinear setting. We are actively working on a nonlinear follow up, as foreshadowed in the discussion section of our manuscript, work that would not have been possible without first studying the properties of this linear model.
>
> In summary, we argue that linear SEMs are both valuable in their own right and useful as a step toward modeling more complex, nonlinear systems.
>
> References:
>
> -https://link.springer.com/book/10.1007/978-1-4612-2748-9
>
> -https://www.jstor.org/stable/2006669
>
> -https://www.jstor.org/stable/2692164
>
> -https://www.tandfonline.com/doi/abs/10.1080/01621459.1995.10476535

---

### Official Review · Reviewer_7aia · 2024-03-15

**Q2-1 Originality-Novelty:** 2
**Q2-2 Correctness-Technical Quality:** 3
**Q2-5 Clarity Of Writing:** 4

**Q10 Ethical Concerns:**

No.

**Q1 Summary And Contributions:**

The paper considers partial identification (bounds, sensitivity intervals) on parameters that are identifiable through Instrumental Variables (e.g., Mendelian randomization) but one crucial assumption is only approximately true, where "approximately is quantified through a sensitivity parameter. (Technically, their A3:exclusion is weakened by merely bounding gamma in their (2) when under exclusion it would be 0.)

Implied bounds on causal effects are a seemingly difficult optimization problem that is solved in near closed form. The bounds can be empty because the model as a whole is falsifiable. The proposed methods seem to work well.

**Q2-3 Extent To Which Claims Are Supported By Evidence:**

4: Excellent: all claims are supported by very convincing evidence (in the form of comprehensive experimental evaluation, rigorous mathematical proofs, detailed (pseudo-)code, precise references, well-motivated and realistic assumptions) and the authors deliver what they promise.

**Q2-4 Reproducibility:**

4: Excellent: key resources (e.g. proofs, code, data) are available and key details (e.g. proof sketches, experimental setup) are comprehensively described for competent researchers to confidently and easily reproduce the main results.

**Q3 Main Strengths:**

The paper tackles an interesting problem, and the essentially closed form solution for low dimensional cases is of interest. I would also agree that the paper tackles a natural generalization of IV; however, that's also a limitation because the concern that the generalization is too natural to not have been considered before holds true (see below). Without this precedent, I would find the paper extremely interesting, and I still think the authords might have something interesting to say, but they need to better carve out the contribution.

**Q4 Main Weakness:**

The main limitation is that the fundamental idea, i.e. to explore bounds induced by relaxing the "exclusion restriction," is anticipated in Conley, Hansen, and Rossi, "Plausibly Exogenous," Review of Economics and Statistics 2012. This is easily ascertained by inspecting the first page here: https://www.jstor.org/stable/41349174, down to the essentially same notation (in particular, gamma is the same in both). It does seem to me that the present paper has some interesting new insight, but the relation must be clarified. In particular, are computational limitations in somewhat high dimensional instances overcome? The authors seem not to claim that.

**Q5 Detailed Comments To The Authors:**

My by far main comment is relation to Conley, Hansen, and Rossi, REStat 2012. Smaller comments follows.

While less close, Nevo and Rosen REStat 2008(ish) might also be worht mentioning.

I agree with your Thm 2 but the proof is very informal. Also, you seem to be worried about the non-bootstrappability of order statistics, but that is quite obviously a wrong analogy? It seems to me that your bounds minimize a smooth and consistently estimated objective, so the problem is analogous to well-behaved m-estimation, and validity of the bootstrap is well understood. That said, there is a small catrch: If the argmins in Thm 1 are close to the boundary of the constraint sets, the bootstrap may not in fact be asymptotically valid or a good finite-sample approximation. Intuitively, you need a CLT for the estimators to have "kicked in."

Also, this seems to be essentially a moment inequalities problem and then we know quite a bit about inference (see Molinari, Handbook of Econometrics 2021). For example, if you want to cover the true theta* whether it is theta+ or theta- but the interval is long relative to standard errors, you can replace alpha/2 with alpha in definitions. On the other hand, since the model is refutable, the CI can in principle be empty and, more worryingly, be misleadingly short if one is close to that scenario. See Andrews and Kwon (REStud 2024) and references therein.

**Q9 Complying With Reviewing Instructions:**

Yes

---

> ### Author Rebuttal · Authors · 2024-04-03
>
> Many thanks to 7aia (henceforth R4) for alerting us to a number of highly relevant publications. We admit we were previously unaware of these works. However, after reading them closely, we find several important differences between the methods they propose and our approach. We have added these references to our discussion.
>
> The Nevo & Rosen paper is primarily focused on the no confounding assumption (A2), rather than the exclusion restriction (A3). As noted in our reply to reviewer d4rf (R1), these two assumptions are often lumped together under the banner of “exogeneity” (see, e.g. Wooldridge 2013, Ch. 15); however, we have made a conscious choice to focus only on the latter in this manuscript. Future work will investigate jointly relaxing (A2) and (A3).
>
> Conley et al. (2012) consider several kinds of prior information on leaky coefficients $\gamma$, including (a) a codomain restriction and (b) a (non-uniform) prior distribution. For (a), they propose estimating a confidence interval for the ATE $\theta$ by taking the union of confidence intervals that result from evaluating a grid of possible values for $\gamma$. This method is computationally demanding (scales exponentially with $d_{Z}$) and likely to produce overly conservative bounds. Moreover, the coverage obtained by this method is a function of the resolution of the aforementioned grid—an arbitrary choice of the user. In particular, the coverage will tend to approach 100% as the grid becomes finer (at least without multiple testing adjustments, which go unmentioned in the paper). By contrast, our convex optimization approach is an efficient one-shot procedure that provides provably sharp bounds (in closed form, for the $L_2$ case). Inference remains potentially challenging for large $d_{Z}$, but this difficulty is outsourced to the covariance matrix estimator. As noted in the text, numerous methods work well in this setting (e.g., regularized or Bayesian approaches), and these can be plugged into our procedure as a modular subroutine.
>
> Conley et al.’s Bayesian variant (b) is intriguing, and we have updated our experiments to include a Bayesian baseline that is closer to this proposal, albeit with a uniform prior (see link below). We note a fundamental challenge with Bayesian approaches to partial identification: since the likelihood is flat within the feasible region, the prior distribution for all parameters plays a decisive role in estimating the posterior distribution. This is appropriate when prior information is rich, but potentially misleading when plugging in default uninformative priors, as some “objective” Bayesians advocate. Notice that the bounds themselves are identifiable by construction (from the population distribution), and statistical inference on the bounds neatly separates estimation error from identification barriers. The standard Bayesian approach does not decouple bounds on the causal parameter from the causal parameter itself, and any claims that the parameter posterior quantiles can be used as “bounds” on the causal effects are either i) a direct consequence of the prior; or ii) heuristics that may be impossible to interpret outside the infinite data limit with an uninformative prior, the latter of which defeats the point of having priors on parameters as a means of tackling unidentifiability.
>
> We also highlight that our revised manuscript elucidates aspects of the leaky IV setting that go un-analyzed by Conley et al. Specifically, in our new Thm. 1, we show that minimum ($\check{\tau}_p$) and oracle ($\tau^*_p$) leakage thresholds fully characterize identifiability conditions in this model. A corollary establishes the link to 2SLS estimation, which is a special case of our leaky model with $\tau = 0$.
> Finally, we have amended the proof for Thm. 2 to streamline the presentation (no discussion of order statistics), clarify the conditional nature of the sampling distribution (only defined within the feasible region), and engage with the moment inequality literature.
>
> References:
>
> -https://cbpbu.ac.in/userfiles/file/2020/STUDY_MAT/ECO/2.pdf
>
> Extra results/figures: https://leakyiv.tiiny.site

---

### Official Review · Reviewer_gn7h · 2024-03-17

**Q2-1 Originality-Novelty:** 2
**Q2-2 Correctness-Technical Quality:** 3
**Q2-5 Clarity Of Writing:** 3

**Q1 Summary And Contributions:**

This paper introduces a novel approach to estimating causal effects using instrumental variables (IVs) that may not fully comply with the traditional exclusion criterion, addressing a common issue in causal inference. The key contribution is a method that offers partial identification in linear models with "leaky instruments," using a convex optimization objective to derive bounds on the average treatment effect. Additionally, it employs a nonparametric bootstrap to evaluate estimate uncertainty.

**Q2-3 Extent To Which Claims Are Supported By Evidence:**

2: Fair: the main claims are somewhat supported by evidence (but the experimental evaluation may be weak, or does not match entirely with the claims, important baselines may be missing, proofs contain important ideas but lack rigor, algorithmic details are only discussed superficially, references are imprecise, assumptions are not sufficiently motivated or explicated, etc.).

**Q2-4 Reproducibility:**

3: Good: key resources (e.g. proofs, code, data) are available and key details (e.g. proofs, experimental setup) are sufficiently well-described for competent researchers to confidently reproduce the main results.

**Q3 Main Strengths:**

1. Focusing on instrumental variables that don't meet the exclusion criteria is interesting but also trivial regarding the literature.
2. The paper is well-written and easy to follow.

**Q4 Main Weakness:**

The key weaknesses are summarized below and will be explored in detail in the next question.

1. Assumption of the linearity of the models.
2. Not a clear comparison with existing methods.
3. Assumption of knowing the population covariance matrix.

**Q5 Detailed Comments To The Authors:**

1. The method relies on linear models, simplifying result derivation but restricting its use in scenarios with non-linear causal relationships. Although the paper briefly explores the general case using mutual information—an intriguing approach—there is room for further development in this area to enhance its applicability.
2. The paper lacks a clear comparison with existing methodologies. Specifically, could the authors provide a more detailed elaboration on Figure 5?
3. The paper assumes knowledge of the population covariance matrix. Although there is a discussion on its estimation, could the authors delve into the sensitivity analysis of the bound results to errors in this estimation? Understanding how estimation inaccuracies impact the bounds could provide insights into the robustness of the proposed method.

**Q9 Complying With Reviewing Instructions:**

Yes

---

> ### Author Rebuttal · Authors · 2024-04-03
>
> Many thanks to reviewer gn7h (henceforth R3) for their valued feedback. We reply to each point in turn.
>
> **Linearity**. We provide a fuller defense of the linearity assumption in our rebuttal to reviewer HFN4. Broadly, we make two points: (1) Many real-world systems are well-described by linear SEMs, which goes some way toward explaining their enormous popularity in econometrics and biology. (2) Studying a linear system is an important step toward understanding more complex nonlinear systems. Thus while linear SEMs are clearly not universal, they are an important topic of study.
>
> **Benchmarks**. We run a series of benchmark experiments against two standard methods for causal effect estimation in linear SEMs (backdoor adjustment and two stage least squares), as well as two more recent methods specifically designed for the case where the exclusion restriction fails (sisVIVE and MBE). All four methods provide point estimates for the ATE, as opposed to our bounding approach. Results are summarized in Fig. 5 (see revised version in the link below). We have also added a new benchmark experiment against a Bayesian alternative, with priors for all model parameters (also in the link below). We find that LeakyIV bounds contain the true ATE in 683 out of 684 settings, or 99.85% of the time. Our bounds are clearly *correlated* with results from IV point estimators, but whereas competitors tend to overstate their confidence – bouncing between positive and negative causal effects multiple times in each panel – our bounds almost never stray so far as to miss the true ATE. In the Bayesian experiment, we find that the posterior tends to concentrate around a biased estimate, occasionally even placing some of the density outside our partial identification interval. Bayesian methods struggle in this setting because every solution in the feasible region has the same likelihood, which makes posteriors especially sensitive to the choice of prior distribution.
>
> **Population covariance**. We do not assume access to the population covariance matrix in our experiments. This was only done in the theory section to clarify what is attainable with perfect information, as is standard in the causality literature on identification, where theoretical results often assume access to a conditional independence oracle (Spirtes et al., 2000). The accuracy of ATE bounds depends on the accuracy of covariance parameter estimates, which in turn depends on the sample size, data dimensionality, and estimator. In Fig. 4B, we show how increasing dimensionality at fixed sample size tends to increase the variance (but not the bias) of our bounds. We have also added a supplemental experiment (not included in our link) demonstrating that our bootstrap confidence intervals attain nominal coverage. These experiments and Thm. 2 on bootstrap inference would be unnecessary with access to the population covariance matrix.
>
> References:
>
> -https://link.springer.com/book/10.1007/978-1-4612-2748-9
>
> Extra results/figures: https://leakyiv.tiiny.site

---

### Official Review · Reviewer_rQbf · 2024-03-21

**Q2-1 Originality-Novelty:** 3
**Q2-2 Correctness-Technical Quality:** 4
**Q2-5 Clarity Of Writing:** 4

**Q1 Summary And Contributions:**

The authors consider the problem of estimating a causal effect in a linear SEM using instruments that violate the exogeneity assumption, that is, affect the outcome directly. They show that if one is willing to assume that the exogeneity violation is bounded one can use the instruments to bound the causal of effect and provide an explicit optimization procedure to compute this bound if the true model covariance matrix is known. They also provide a bootstrap procedure to compute confidence intervals for the bound in the more practically relevant setting where the model covariance matrix has to be estimated.

**Q2-3 Extent To Which Claims Are Supported By Evidence:**

4: Excellent: all claims are supported by very convincing evidence (in the form of comprehensive experimental evaluation, rigorous mathematical proofs, detailed (pseudo-)code, precise references, well-motivated and realistic assumptions) and the authors deliver what they promise.

**Q2-4 Reproducibility:**

4: Excellent: key resources (e.g. proofs, code, data) are available and key details (e.g. proof sketches, experimental setup) are comprehensively described for competent researchers to confidently and easily reproduce the main results.

**Q3 Main Strengths:**

Instrumental variables are extremely popular in wide array of areas including econometrics and medicine. In most practical use cases of IV, the instruments used are likely to violate the exogeneity assumption and it is therefore very natural to investigate this setting. The paper considers this problem in a simple setting (unconditional IV, no violation of the no confounding assumption, linear SEM) but does so in a concise and technically elegant manner. The paper is also very well written and easy to follow.

**Q4 Main Weakness:**

The paper considers quite a specific class of models (unconditional IV, instrument parent of treatment, no violation of the no-confounding assumption for the instrument, linear SEM) which limits the relevance of the results. The tools used to derive the results also seem unlikely to remain sufficient to tackle the problem if one were to drop these assumptions. Further, the paper seems at times to implicitly assume Gaussianity in it's presentation (although I don't think this is relevant for the results). For example, they use the term conditional variance even though the definition they give is only the conditional variance if the variables are Gaussian. Similarly, the definition of a valid instrument in the second paragraph of Section 2.2 is only true with caveats in the general case (see for example Saengkyongam et al., 2022) but is true for linear Gaussian SEMs where the conditional independences may be replaced with covariances. In general this paragraph is lacking in rigor and references, although this is a reflection of the general IV literature.

**Q5 Detailed Comments To The Authors:**

- Corollary 1.1: I suppose b \cdot b = b^t b? Why the change of notation compare to for example the definition of theta_2 in the paragraph at the bottom left of page 4.
- Could you discuss more thoroughly what it means that in some instance there is not average treatment affect compatible with the data in Theorem 1? Does this imply that the constraint imposed by the user on the leaky instrument is too strong or could this be used to detect whether the instrument is valid? How should the user proceed if they encounter such a case?

**Q9 Complying With Reviewing Instructions:**

Yes

---

> ### Author Rebuttal · Authors · 2024-04-03
>
> Many thanks to rQbf (henceforth R2) for their largely positive review. R2 rightly points out a notational inconsistency in Corollary 1.1 that has now been amended. In reply to R2’s second question, we have expanded the manuscript’s discussion of identifiability, culminating in a new Thm. 1 that characterizes identifiability conditions in the leaky IV model (see link below). Strictly speaking, any ATE value is compatible with the data under sufficient confounding. However, some degrees of information leakage are incompatible with the data – specifically, any leakage less than $\check{\tau}_p$. Since this parameter can be estimated directly, we can falsify some overly optimistic assumptions about the true information leakage; however, thresholds in between the minimum and oracle values will incur errors. For more details, see the link below.
>
> R2 raises questions about implicit normality assumptions arising from our interpretation of $\kappa$ parameters as conditional (co)variances. In fact, this follows directly from the linear structural equations for $X$ and $Y$ (Eqs. 1-2) and requires no distributional assumptions (aside from finite variance, which we have now made explicit). It can be verified that our $\kappa$s are the conditional (co)variances by starting from the definition of e.g., Cov$(X, Y \mid Z)$, and substituting in our structural equations.
>
> We are not sure we fully understand R2’s comment on the definition of IVs provided by (A1)-(A3). This is a classic nonparametric formulation (see, e.g., Balke & Pearl, 1997). We do not intend to “replace” conditional (in)dependence claims with conditional covariance claims *in general*. However, in the linear SEM defined by Eq. 1-3, we can equate particular values of model parameters with particular conditional (in)dependence relations. For instance, (A1) and Eq. 1 jointly hold iff $\lVert \beta \rVert > 0$, while (A3) and Eq. 2 jointly hold iff $\lVert \gamma \rVert = 0$. We will revise this section to better clarify this and add a reference to Saengkyongam et al. (2022).
>
> References:
>
> -https://www.jstor.org/stable/2965583
>
> Extra results/figures: https://leakyiv.tiiny.site

---

### Official Review · Reviewer_d4rf · 2024-03-22

**Q2-1 Originality-Novelty:** 3
**Q2-2 Correctness-Technical Quality:** 2
**Q2-5 Clarity Of Writing:** 3

**Q1 Summary And Contributions:**

The authors introduce a method for bounding the causal effect by some invalid instrumental variables, termed as ``leaky instruments", within a linear structural equation model. They demonstrate the bounds are sharp and provide a generic bootstrapping method that ensures reliable coverage for the estimated bounds.

**Q2-3 Extent To Which Claims Are Supported By Evidence:**

2: Fair: the main claims are somewhat supported by evidence (but the experimental evaluation may be weak, or does not match entirely with the claims, important baselines may be missing, proofs contain important ideas but lack rigor, algorithmic details are only discussed superficially, references are imprecise, assumptions are not sufficiently motivated or explicated, etc.).

**Q2-4 Reproducibility:**

3: Good: key resources (e.g. proofs, code, data) are available and key details (e.g. proofs, experimental setup) are sufficiently well-described for competent researchers to confidently reproduce the main results.

**Q3 Main Strengths:**

The notion of leaky instruments is new and the estimating method is novel.

The authors thoroughly explore the limitations of their proposed technique and outline potential strategies for overcoming these challenges.

**Q4 Main Weakness:**

The authors have confined their model to scenarios that contravene the A3 criterion. However, this limitation might exclude a significant portion of real-world data from being effectively applied within the model's framework. It's often challenging in practical applications to ascertain the presence of a relationship between instrumental variables and unobserved confounders.

Regarding the experimental section, it would greatly enhance the paper's applicability and relevance if the authors could incorporate an example using real data. This addition would help demonstrate the model's practical utility and its performance in real-world scenarios.

**Q5 Detailed Comments To The Authors:**

Is it necessary for the causal effect $\beta$ of the instrumental variable
$Z$ on the explanatory variable
$X$ to be more substantial than the causal effect  of $\gamma$ of $Z$ on $Y$? To put it differently, does the weakness of the instrumental variable
$Z$ compromise the accuracy of estimating the bounds of the causal effect?

Furthermore, the paper discusses that the causal effect $\theta$ can be represented through a parameter $\tau$. Could the authors elucidate this transformation process in a more intuitive manner? Such a clarification would significantly enhance the reader's understanding of the underlying mechanisms and assumptions.

**Q9 Complying With Reviewing Instructions:**

Yes

---

> ### Author Rebuttal · Authors · 2024-04-02
>
> Thanks to reviewer d4rf (henceforth R1) for their constructive feedback. R1 raises a valid point about violations of the no confounding assumption, which we label (A2) in our manuscript. We are currently working on a follow up project in which we tackle both (A2) and (A3) jointly – and indeed, the two are quite intertwined. (So much so, in fact, that the term “exogeneity” in econometrics is often defined so as to cover both; see, e.g. Wooldridge 2013, Ch. 15). However, for matters of clarity and scope, we made a deliberate choice to focus on (A3) in this manuscript, as information leakage between instruments and outcomes is a major concern in many application areas. For instance, in Mendelian randomization studies, horizontal pleiotropy is an (A3) violation that may severely bias results (Hemani et al., 2018).
>
> Our focus on simulated data is motivated by the need to demonstrate performance across a range of values for the signal-to-noise ratios of models for $X$ and $Y$, as well as oracle leakage $\lVert \gamma^* \rVert_2$, parameters which cannot be observed in real world settings. In our extended follow up paper, we will focus on applications in econometrics and population genetics.
>
> R1 asks about the performance of LeakyIV with weak instruments. There is no formal requirement that instruments be more “relevant” than “leaky” (i.e., $\lVert \beta \rVert_2 \geq \lVert \gamma \rVert_2$, assuming standardized variables throughout) but strong instruments will tend to produce shorter ATE intervals, all else being equal. (This follows from the definition of function $g$ in Lemma 2.) Thus, it remains the case that if practitioners want more information about the value of $\theta$, they are better served by strong instruments than weak ones, even when point identification is impossible.
>
> R1 requests more information about the relationship between the ATE $\theta$ and the leakage threshold $\tau$. Recall that $\theta$ is related to overall leakage $\lVert \gamma \rVert_p$ by the function $g_p(\theta) := \lVert \alpha - \theta \beta \rVert_p$, where $\alpha := \Sigma_{zz}^{-1} \cdot \Sigma_{zy}$ and $\beta := \Sigma_{zz}^{-1} \cdot \Sigma_{zy}$ represent the expected coefficients of an OLS regression of $Y$ on $Z$ and $X$ on $Z$, respectively (see Lemmas 2 and 3, which we have combined into a single Lemma in our revised manuscript). The user-defined threshold parameter $\tau$ places an upper bound on this norm, and so restricts our identification task to the space of models satisfying $g_p(\theta) \leq \tau$. All of this is made more explicit in our revised Section 3, where we have also added a new theorem on identifiability (see link below)
>
> References:
>
> -https://cbpbu.ac.in/userfiles/file/2020/STUDY_MAT/ECO/2.pdf
>
> -https://academic.oup.com/hmg/article/27/R2/R195/4996734
>
> Extra results/figures: https://leakyiv.tiiny.site

---

### Meta-Review · Area_Chair_ahjg · 2024-04-15

The paper introduces a procedure to bound causal effects in a linear instrumental variable setting with invalid instruments. Overall the reviewers assessed the submission positively and consider the addressed problem interesting and believe the work offers a valid technical contribution. At the same time they also pointed out that some important related work,  in which a similar bounding approach is proposed, had been missed by the authors. In their rebuttal the authors acknowledge that the reference was missed, but argued that their proposal is different. After the discussion phase it appears that there is agreement that if the missed reference is mentioned (together with a short discussion on how the current approach differs) the current submission still makes a valuable contribution.